Corrected: Author correction

# Lineage tracing of acute myeloid leukemia reveals the impact of hypomethylating agents on chemoresistance selection

Francisco Caiado [1*], Diogo Maia-Silva [1,2], Carolina Jardim[1], Nina Schmolka [1,3], Tânia Carvalho[1], Cláudia Reforço [1], Rita Faria [1], Branka Kolundzija[1], André E. Simões[1], Tuncay Baubec [3], Christopher R. Vakoc[2], Maria Gomes da Silva[4], Markus G. Manz [5], Ton N. Schumacher [6], Håkan Norell[1,7*] & Bruno Silva-Santos [1,7*]

Chemotherapy-resistant cancer recurrence is a major cause of mortality. In acute myeloid leukemia (AML), chemorefractory relapses result from the complex interplay between altered genetic, epigenetic and transcriptional states in leukemic cells. Here, we develop an experimental model system using in vitro lineage tracing coupled with exome, transcriptome and in vivo functional readouts to assess the AML population dynamics and associated molecular determinants underpinning chemoresistance development. We find that combining standard chemotherapeutic regimens with low doses of DNA methyltransferase inhibitors (DNMTi, hypomethylating drugs) prevents chemoresistant relapses. Mechanistically, DNMTi suppresses the outgrowth of a pre-determined set of chemoresistant AML clones with stemness properties, instead favoring the expansion of rarer and unfit chemosensitive clones. Importantly, we confirm the capacity of DNMTi combination to suppress stemness-dependent chemoresistance development in xenotransplantation models and primary AML patient samples. Together, these results support the potential of DNMTi combination treatment to circumvent the development of chemorefractory AML relapses.

[1] Instituto de Medicina Molecular João Lobo Antunes, Faculdade de Medicina, Universidade de Lisboa, Lisboa, Portugal. [2] Cold Spring Harbor Laboratory, Cold Spring Harbor, New York, NY, USA. [3] Department of Molecular Mechanisms of Disease, University of Zurich, Zurich, Switzerland. [4] Instituto Portugues de Oncologia—Francisco Gentil, Lisbon, Portugal. [5] Department of Medical Oncology and Hematology, University Hospital Zurich and University of Zurich, Zürich, Switzerland. [6] Netherlands Cancer Institute, Amsterdam, The Netherlands. [7] These authors jointly supervised this work: Håkan Norell, Bruno Silva-Santos. *email: caiado12@gmail.com; haakannorell@gmail.com; bssantos@medicina.ulisboa.pt

Chemotherapy resistance (chemoresistance) is a major driver of cancer recurrence[1]. Intra-tumor heterogeneity (ITH), the end product of co-existing microenvironmental, phenotypical, transcriptomic, epigenetic, and genetic variants, drives chemoresistance by providing multiple substrates for tumor escape under the selective pressure of chemotherapy[2]. Genetic ITH results from continuous cycles of mutation, selection and expansion under context-specific selective pressures—i.e. cancer clonal evolution—and contributes to AML chemorefractory relapses through expansion of sub-clonal population(s) harboring either intrinsic (pre-existing) or acquired (therapy-induced) chemoresistance-promoting mutations[1,3]. In addition to genetic ITH, non-genetic ITH is also a major contributor to chemoresistance development. Heterogeneous epigenetic regulation of gene expression has been shown to generate hierarchally related but phenotypically divergent co-existing cell subpopulations originating from genetically identical tumor cells[4]. A clinically relevant example of non-genetic hierarchical organization of some tumors are cancer stem cells (CSC), which are genetically identical to the bulk of the tumor, but display substantially higher tumorigenic capacity than their isogenic siblings. CSCs have been implicated in chemoresistance and recurrence in various cancer types, which arise as a result of their unique properties. These are often encompassed in the term stemness, and include slow cell cycle progression (or quiescence), upregulation of drug-efflux pumps, protection from reactive oxygen species, high self-renewal or tumor initiation capacity in immunocompromised mice[5].

Development of chemoresistant relapses is of particular importance in acute myeloid leukemia (AML), making this the deadliest blood cancer[6]. AML relapse to standard chemotherapy has been traced to pre-existing genetically defined clones that acquire additional mutations, evolving into the dominant relapsing sub-clones[7–10]. Despite the established role of specific genetic alterations in diagnosis, prognosis and treatment stratification[11–14], AML is a highly heterogeneous disease with surprisingly lower average number of mutations than most other adult cancers[15,16], suggesting that non-genetic factors are also relevant in AML outcomes. In fact, it was shown that AML relapse to standard chemotherapy depends heavily on transcriptional stemness programs[17–19]. The pervasiveness of non-genetic ITH is also evidenced by the existence of extensive epigenetic alterations in AML genomes. For example DNA methylation, an epigenetic modification that impacts transcription, carries diagnostic and prognostic value in AML[20–22], with recent studies establishing DNA hypermethylation as a poor prognosis factor in de novo AML[23]. In contrast to genetic changes, epigenetic modifications are frequently reversible, which provides opportunities for their reversion to non-pathogenic states by the use of specific inhibitors. DNA methyltransferase inhibitors (DNMTi) or hypomethylating agents like decitabine (DAC) and azacitadine (Aza) have established clinical benefits in myelodysplastic syndrome (MDS) patients and are also approved as monotherapy for certain groups of elderly AML patients[24]. Importantly, different clinical studies combining low doses of DAC with standard chemotherapy regimens have shown clinical benefit, particularly in patients with refractory/relapsed AML[25–27]. In spite of its clinical benefit, the impact of combining DNMTis with chemotherapeutic regimens on the different layers of ITH and its overall effect on de novo AML chemoresistance development remains largely unexplored, mostly due to the lack of adequate experimental systems. A key approach to assess the impact of different therapies on ITH is the use of lineage-tracing technologies. Recent in vitro lineage-tracing studies have revealed the pre-existing nature of targeted therapy resistance in lung cancer and chronic myeloid leukemia models[28], thus attesting the validity of in vitro lineage tracing experimental systems to dissect the consequences of ITH on cancer biology.

Here, we employ lineage tracing (DNA barcodes—BCs) to assess the longitudinal clonal dynamics, beyond the level of genetics, that underlie the emergence of chemoresistance in human AML (hAML) cells in the presence or absence of DNMTi. To assess the contribution of different layers of ITH, we combine lineage tracing with exomic, transcriptomic and phenotypic profiling together with in vivo functional assessment of hAML cells relapsing to chemotherapy. Using this approach, we find that standard chemotherapy drives the selection of a pre-determined set of recurring AML clones with increased in vivo leukemia-initiating capacity and resistance to a second round of chemotherapy. Strikingly, we reveal that low-dose DNMTi in combination with chemotherapy selects for an alternative unpredictable set of clones with decreased stemness properties that remain sensitive to chemotherapy. Collectively, our findings attest the potential of a combinatorial approach of standard chemotherapy with low-dose DNMTi to circumvent chemoresistance development during AML treatment, namely through marked shaping of the underlying clonal dynamics and transcriptomic landscape.

## Results

**Chemotherapy + DAC combination prevents hAML chemoresistance.** In order to model the development of chemoresistance in hAML cells and assess its underlying clonal dynamics, we established an in vitro system using barcoded hAML cell lines (HEL and OCI-AML3) and exposed them to standard chemotherapeutic regimens optimized in order to significantly deplete viable cells (over 99% elimination). Additionally, to model disease recurrences observed in chemotherapy-treated AML patients, we tested chemotherapeutic regimen doses that allowed barcoded hAML cell regrowth post-therapy exposure, thus generating in vitro relapses (Fig. 1a). A key aspect of this system is that lineage-tracing using DNA barcodes (BCs) allows the quantitative tracking of cell populations derived from each initially barcoded single-cell[29], defined as barcoded clones (BC-clones), thus allowing quantification of BC-clonal dynamics in response to chemotherapeutic regimens. Barcoded hAML cells lines were exposed for 72 h to regimens of standard chemotherapy (doxorubicin–Doxo: 1.8 μM and cytarabine–Cyta: 6 μM) alone or simultaneously combined with a low-dose of hypomethylating agent decitabine (DAC 0.1 μM). Viable cell numbers were monitored from treatment initiation (T0) until the time point when post-therapy cell cultures reached equivalent numbers as T0, termed Trelapse. We observed that Doxo, Doxo + Cyta and Doxo + Cyta + DAC regimens strongly depleted barcoded hAML cells in both cell lines (for example, depletion of 99.56%(±0.23% eliminated cells, $n = 6$), 99.96% (± 0.020%, $n = 8$) and 99.90%(±0.070%, $n = 8$) HEL cells, respectively), whereas DAC single treatment had minor effects on cell numbers (Fig. 1b). Despite the high levels of cell elimination, Doxo ±Cyta and Doxo + Cyta + DAC treated cells were able to re-grow back to the initial cell number within 30–35 days after treatment initiation. Next, to assess cell-intrinsic gain of chemoresistance in Trelapse cell populations, we re-exposed Trelapse AML cells to chemotherapy (Doxo + Cyta) for 72 h. Strikingly, whereas chemo-exposed (Doxo±Cyta) AML cells showed gain of resistance, upfront co-exposure to low-dose DAC preserved sensitivity to chemotherapy re-treatment, similarly to the no-treament (NT) control group (Fig. 1c). In agreement, Doxo±Cyta Trelapse samples showed a significant (2–4-fold) increase in IC50 values for doxorubicin compared to either NT or Doxo + Cyta + DAC relapse samples (Table 1). To confirm the capacity of upfront DAC combination to prevent chemoresistance development, we re-treated Trelapse samples with a new round of chemotherapy and quantified the frequency of Trelapse samples (Doxo, Doxo + Cyta and Doxo + Cyta + DAC) that generated a second relapse. We observed that

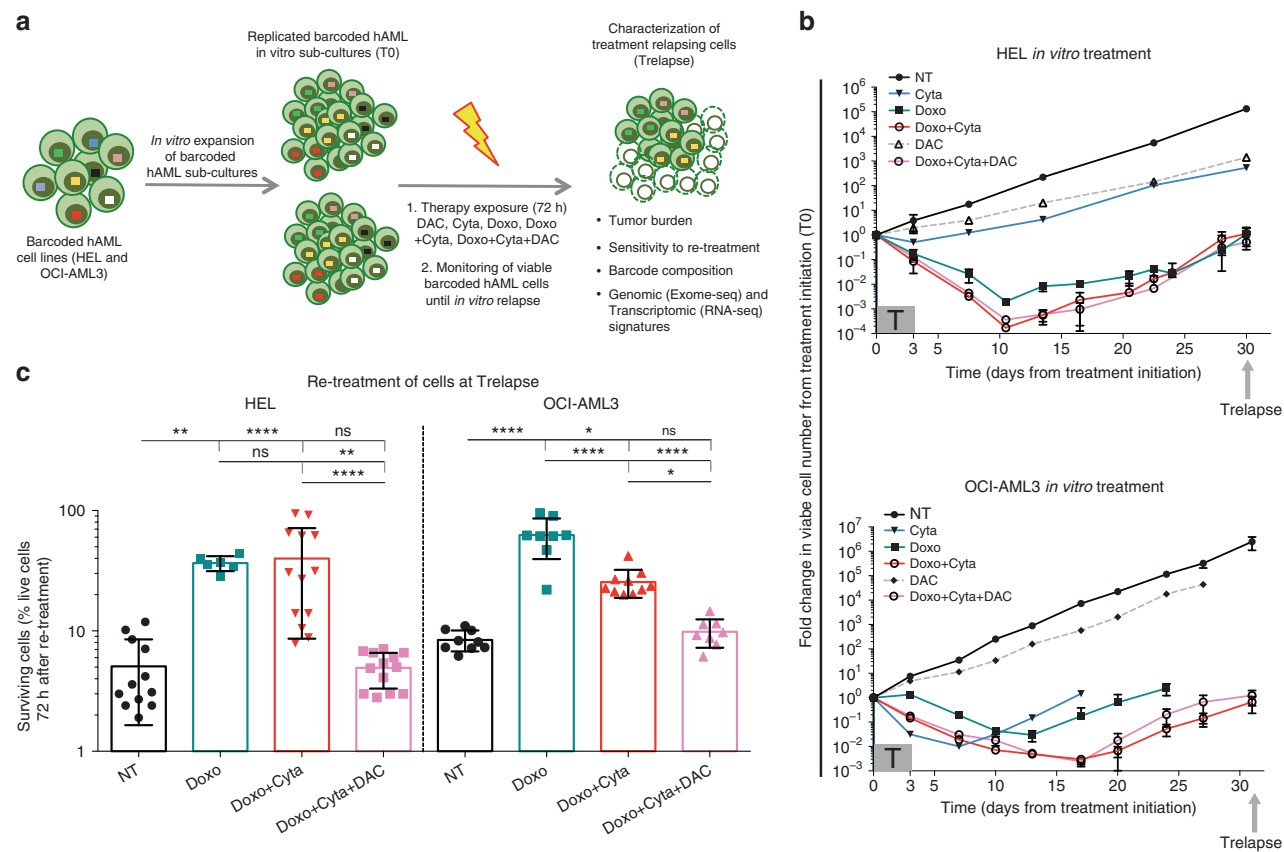

**Fig. 1** Chemotherapy + Decitabine combination prevents chemoresistance in hAML cells. **a** Schematic diagram of the experimental system used. T0 indicates the time of treatment initiation and Trelapse is defined as the post-therapy time point when re-grown hAML reached equivalent numbers as T0. NT–no treatment; DAC–decitabine (0.1 μM); Doxo–doxorubicin (1.8 μM); Cyta–cytarabine (6μM). **b** Total number of viable HEL and OCI-AML3 barcoded cells between T0 and Trelapse, defined as a fold variation of cell number at each measured time point relatively to T0 ($n = 3$ for NT and DAC, $n = 5$ for Cyta and Doxo, $n = 8$ for Doxo + Cyta and Doxo + Cyta + DAC, independent replicates). T box indicates the period of therapy exposure. **c** Frequency of viable Trelapse HEL and OCI-AML3 barcoded cells (NT: $n = 12/9$, Doxo: $n = 6/8$, Doxo + Cyta: 13/10 and Doxo + Cyta + DAC: 13/8, independent replicates in HEL / OCI-AML3 respectively) after re-exposure to Doxo + Cyta for 72 h. Concerning panels b and c displayed graphs show mean ± s.d.; $P$ values were determined by one-way ANOVA test. ns–not significant, *$P < 0.05$; **$P < 0.01$; ***$P < 0.001$; ****$P < 0.0001$. Source data are provided as a Source Data file

| Table 1 Doxorubicin IC50 values of Trelapse hAML cells | | | | | |
|---|---|---|---|---|---|
| | | **Trelapse Samples** | | | |
| | **Cell line** | **NT** | **Doxo** | **Doxo + Cyta** | **Doxo + Cyta + DAC** |
| IC50 (μM) ($n = 3$) | HEL | 0.345 ± 0.0352 | 1.40 ± 0.422 | 1.44 ± 0.185 | 0.571 ± 0.0242 |
| | OCI-AML3 | 1.31 ± 0.00503 | 4.20 ± 0.256 | 3.80 ± 0.351 | 1.40 ± 0.335 |
| $P$-value (t-test, relative to NT) | HEL | NA | 0.0125 | 0.0005 | 0.0008 |
| | OCI-AML3 | NA | <0.0001 | 0.0003 | 0.672 |
| IC50 ratio (relative to NT) | HEL | 1 | 4.06 | 4.20 | 1.66 |
| | OCI-AML3 | 1 | 3.20 | 2.90 | 1.07 |
| NA not applicable | | | | | |

while 100% of Doxo±Cyta Trelapse samples relapsed a second time, only 25% (1 out of 4) of Doxo + Cyta + DAC Trelapse samples was able to relapse to re-treatment (Supplementary Fig. 1a, b). Importantly, by comparing the number of live cells at an equivalent time point (11 days) after treatment and re-treatment, we observed that Doxo ± Cyta Trelapse samples were significantly less sensitive to re-treatment, further confirming increased chemoresistance in these groups (Supplementary Fig. 1c). Overall, these results show that in vitro upfront combination of chemotherapy with low-dose DAC prevents the emergence of chemoresistant hAML cells.

**Chemotherapy selects for a pre-determined set of BC-clones.** To explore the clonal dynamics resulting from different treatment regimens we evaluated the BC-clonal composition of T0 and Trelapse samples (Supplementary Fig. 2a–d). In the absence of therapy (NT), we observed stable and highly correlated (pearson correlation coefficient > 0.7) BC-clone frequencies at day 30 relatively to T0 and also between replicates at Trelapse, even after >10⁵-fold expansion (Supplementary Fig. 2c–j). This validates the clonal stability of our system in the absence of therapeutic pressure, thus allowing us to attribute BC clonal variations to therapeutic selection (rather than stochasticity of the system). Among

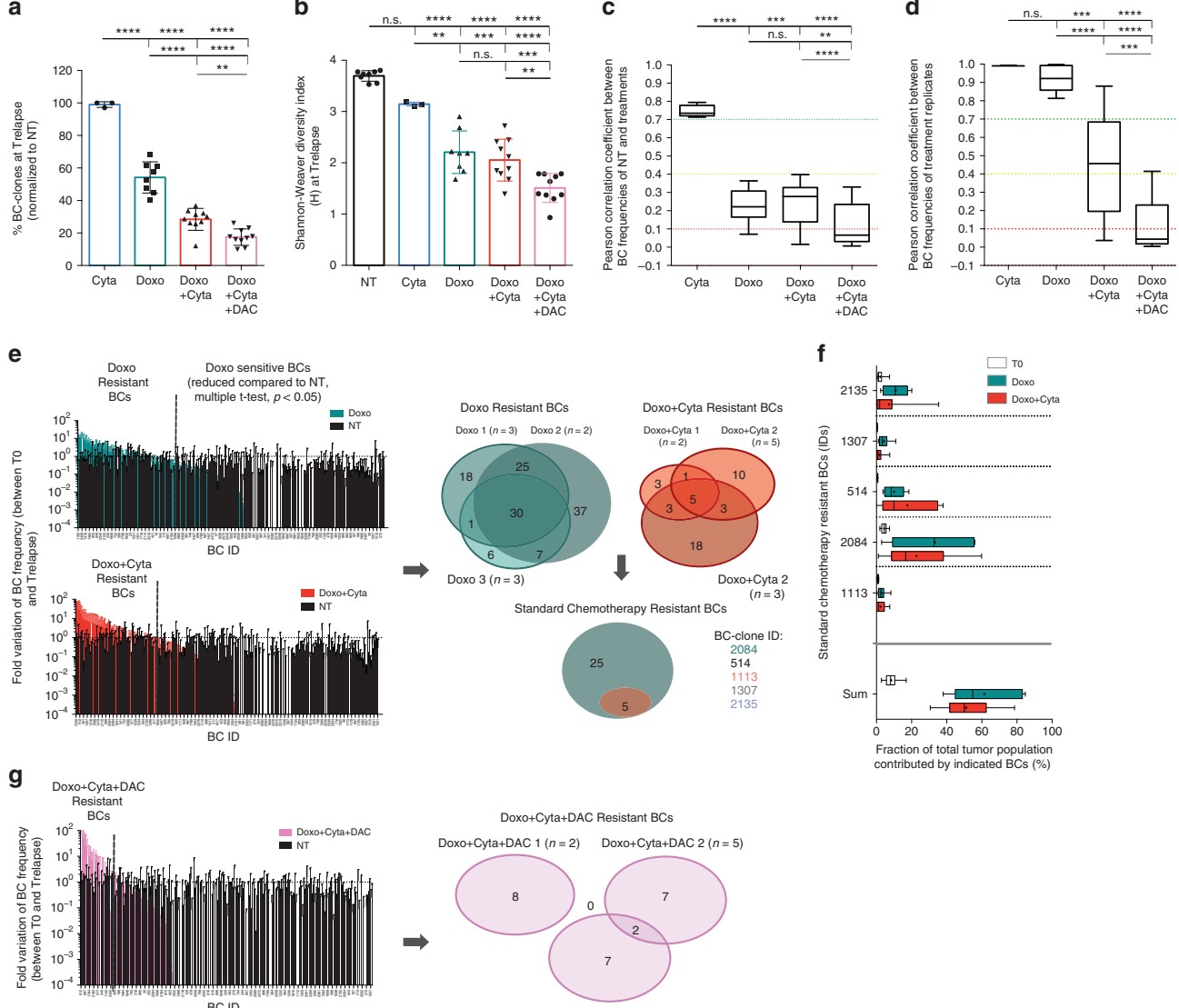

**Fig. 2** Pre-determined set of BC-clones associates with chemoresistant relapses. (Data relative to HEL cell line) **a** Normalized (to NT) frequency of barcodes detected at Trelapse in Cyta ($n = 3$), Doxo ($n = 8$), Doxo + Cyta ($n = 10$) and Doxo + Cyta + DAC ($n = 10$) groups. **b** Shannon-Weaver diversity index ($H$) of indicated Trelapse groups. $H = -\mathrm{Sum}(xi\mathrm{LN}xi)$, where $xi$ is the frequency of each BC-clone in the population. $H$ reflects the BC number and how evenly distributed these are in the population (higher $H$ results from higher BC number and more even distribution). **c** Pearson correlation coefficient between BC-clonal architectures of each treatment and NT groups (Cyta: $n = 9$, Doxo: $n = 19$, Doxo + Cyta ± DAC = 36). **d** Pearson correlation coefficient between BC-clonal architectures of replicates within each treatment group at Trelapse.(Cyta: $n = 3$, Doxo: $n = 10$, Doxo + Cyta ± DAC = 14) **e** Strategy used to define Doxo and Doxo + Cyta resistant BC-clones. On the left are depicted the fold variations of each BC-clone between T0 and Trelapse in Doxo and Doxo + Cyta groups compared to NT, chemoresistant BC-clones were defined has the ones showing equal or increased fold variation relative to NT, determined by preforming multiple $t$-testing (per individual BC, $P < 0.05$). This was performed in three independent experiments ($n = 2$, $n = 5$, $n = 3$). On the right, venn diagrams depicting the overlap of chemoresistant BC-clones, shared between all the replicates of three independent experiments for Doxo and Doxo + Cyta groups; as well as the overlap between these respective cores. The list of the chemoresistant BC-clones is indicated with respective BC-identification (id). **f** Average frequency of each indicated BC-clone (and their summed frequency) in Doxo ($n = 8$), Doxo + Cyta ($n = 10$) at Trelapse and T0 ($n = 5$). Boxplots **c**, **d** and **f**: center line is median, lower box bound is the 25% percentile, upper box bound is the 75% percentile, whiskers are minimum and maximum values. Cross is the mean (**f**). **g** Same as **e** but applied to Doxo + Cyta + DAC group. Graphs of mean ± s.d., $P$ values were determined by one-way ANOVA test. ns–not significant, *$P < 0.05$; **$P < 0.01$; ***$P < 0.001$; ****$P < 0.0001$. Source data are provided as a Source Data file

the Trelapse samples that were significantly impacted by therapy (Doxo, Doxo + Cyta, Doxo + Cyta + DAC), chemosensitive hAML cells relapsing to Doxo + Cyta + DAC combination showed lowest BC numbers and diversity (lowest Shannon-Weaver diversity index H) (Fig. 2a, b, Supplementary Fig. 3a–c) which reflected in clonal architectures most divergent from NT samples (Fig. 2c, Supplementary Fig. 3d). By evaluating correlations between the BC architectures across replicates of each

treatment at Trelapse, we found that BC distributions across Doxo relapses were highly reproducible (pearson > 0.7) while addition of Cyta decreased the similarity of replicates, and further combination with DAC effectively abrogated all correlations (pearson < 0.1) (Fig. 2d, Supplementary Fig. 3e). This suggests that DAC combination poses a stronger selective pressure on the system towards reducing the BC-clonal diversity and leading to relapses mediated by unpredictable (non-shared) BC-clones. To

test if this effect was not the direct result of higher cell elimination (significantly different between Doxo and Doxo + Cytav ± DAC groups), we preformed drug dose titrations that led to different cell elimination levels and assessed the corresponding BC-clone numbers at Trelapse. We observed a positive correlation between the number of live cells (at maximum selection point) and the equivalent number of detected BC-clones in each treatment titration, with the highest correlation coefficient observed in the Doxo + Cyta + DAC group (Supplementary Fig. 4a). Additionally, we compared Doxo + Cyta ± DAC groups with a group receiving 9-fold higher doxorubicin concentration (Doxo9x). We confirmed that under equal cell elimination levels, the Doxo + Cyta + DAC group showed higher BC-clone elimination and, contrarily to the other groups, remained chemosensitive (Supplementary Fig. 4b–d). These data suggest that the level of cell elimination drives BC-clone elimination in all conditions, but DAC combination selectively shows an increased capacity to deplete BC-clones even upon normalization of cell elimination levels. Next, the higher correlation between replicates in Doxo ± Cyta treated samples compared to the Doxo + Cyta + DAC group prompted us to investigate if chemoresistant relapses shared a common set of BC-clones. For this, we analyzed the fold variation of each individual barcode frequency between T0 and Trelapse and based on statistical significance (multiple *t*-test) identified BC-clones with equal or increased frequency in Doxo ± Cyta relapses compared to NT, which were defined as Doxo and Doxo + Cyta resistant BC-clones. By overlapping Doxo ± Cyta resistant BC-clones from three independent experiments, we identified a group of 5 or 6 BC-clones (in HEL and OCI-AML3, respectively) that were consistently selected in chemoresistant relapses (Fig. 2e, Supplementary Fig. 3g); and collectively constituted a major fraction of those Trelapse populations: 58,8% (±19.5% tumor fraction in Doxo/HEL, $n = 8$) and 50%(±10.8% in Doxo + Cyta/HEL, $n = 10$), which represented a 7–9-fold expansion compared to T0 (Fig. 2f). Strikingly, the same analysis on Doxo + Cyta + DAC Trelapse samples revealed that no BC-clone was consistently selected upon this treatment (Fig. 2g) further confirming the unpredictable nature of relapse upon DAC combination. Finally, to confirm the chemoresistance of Doxo ± Cyta-selected BC-clone set, we evaluated its presence in second relapse (R2) samples after re-treatment. We observed that although R2 samples showed reduced BC-clone number compared to first relapse (R1) samples, their BC architecture was highly correlated with R1 (Supplementary Fig. 1d, e), suggesting a conserved BC composition to R1. Furthermore, there was an expansion of Doxo ± Cyta resistant BC-clones summed frequency in R2 relatively to R1 (Supplementary Fig. 1f, g). Altogether, these data indicate that, whereas chemoresistant relapses associate with selection and expansion of a pre-determined set of BC-clones (that persist after two rounds of chemotherapy), DAC combination-driven chemosensitive relapses have reduced BC-clonal diversity and are mediated by unpredictable BC-clones.

**DAC combination selects unpredictable and unfit BC-clones.** The ability of DAC combination to generate chemosensitive hAML relapses, led us to dissect the effects of DAC on hAML clonal dynamics. Concerning the effect of low-dose DAC as a single agent, we observed a mild reduction on genomic DNA methylated cytosine frequency, and a decrease in the population doubling time (Supplementary Fig. 5a, b), as previously described[30]. At the BC level, these cells showed great similarity to NT samples and also between replicates (Supplementary Fig. 5c–e), suggesting that low-dose DAC alone has minimal impact on hAML BC-clonal dynamics. Next, we assessed whether DAC combination with chemotherapy impacted on the selection of the

above-identified pre-determined set of chemoresistant BC-clones. By assessing BC-clonal compositions in the different experimental groups, we observed that, while the pre-determined BC set (with particular dominance of BC-clone 2084; dark green) was enriched in Doxo ± Cyta relapses, it was strikingly under-represented in Doxo + Cyta + DAC relapses (Fig. 3a). In agreement, these showed significantly reduced frequency of chemoresistant BC-clones in the top 3 most frequent BC-clones compared to Doxo±Cyta relapses (Fig. 3b). To improve our assessment of the effect of DAC combination on the pre-determined set of chemoresistant BC-clones, we quantified the frequency fold change of each of these clones between T0 and Trelapse under each treatment, and normalized it to the equivalent fold change observed in NT conditions—hereby defined as competitive index (CI). Strikingly, CI for the majority of the BC-clones was significantly reduced under the selective pressure of Doxo + Cyta + DAC compared to Doxo ± Cyta (Fig. 3c, Supplementary Fig. 3h), suggesting that DAC combination impairs the chemotherapy-driven selection of pre-determined chemoresistant BC-clones. To validate these findings with another hypomethylating agent, we evaluated the combination of chemotherapy with azacitidine (Aza). Like DAC, Aza also prevented chemoresistance while supressing pre-determined chemoresistant BC-clones (Supplementary Fig. 6a–g), strongly supporting the conclusion that this is a general effect of hypomethylating drugs. We next sought to characterize the alternative BC-clones driving chemosensitive relapses upon DAC combination. Given the previously shown lack of a conserved set of BC-clones specifically selected by Doxo + Cyta + DAC, we focused our analysis of top 3 most frequent relapsing BC-clones per culture. We observed that the most abundant BC-clones of Doxo + Cyta + DAC were rarer at T0 but underwent greater expansions compared to chemoresistant BC-clones relapsing after chemotherapy alone (26 versus 7–13 fold, in HEL) (Fig. 3d, Supplementary Fig. 3i). Importantly, by assessing the fold change of the frequency of each individual BC-clone between T0 and Trelapse in NT conditions—hereby defined as fitness—we observed that the top 3 BC-clones of Doxo + Cyta + DAC had reduced fitness compared to those of Doxo ± Cyta (Fig. 3e, Supplementary Fig. 3j), indicating that in the absence of therapy these rarer clones are outcompeted. As expected, the average CI of the top 3 BC-clones relapsing to Doxo + Cyta + DAC was significantly higher compared to those of Doxo ± Cyta, confirming their gain in competitiveness specifically in the context of DAC combination (Fig. 3f). Finally, to assess if DAC addition impaired chemotherapy-driven BC-clone selection preferentially in the elimination or re-growth stages of our in vitro model, we preformed detailed longitudinal assessment of BC-clonal dynamics after treatment (Supplementary Fig. 7a). Focusing on the most dominant chemoresistant BC-clone (2084), we observed that DAC combination had only a mild impact on the initial elimination stage, but strongly impacted on its capacity to re-grow (Supplementary Fig. 7b, c). By contrast, the less fit and rarer BC-clone 2252 was suppressed by Doxo + Cyta (Fig. 7d) but expanded massively and selectively under DAC combination (Supplementary Fig. 7e). Altogether, these data show that DAC combination suppresses the re-growth of chemoresistant clones, while favoring the selection and massive expansion of rarer and less fit clones that remain sensitive to chemotherapy re-treatment.

**DAC shapes the transcriptome and function of hAML relapses.** Having established the impact of DNA hypomethylating agents on hAML clonal dynamics underlying chemoresistance development, we next sought to decipher its molecular basis. In order to clarify the contribution of genetic and transcriptomic

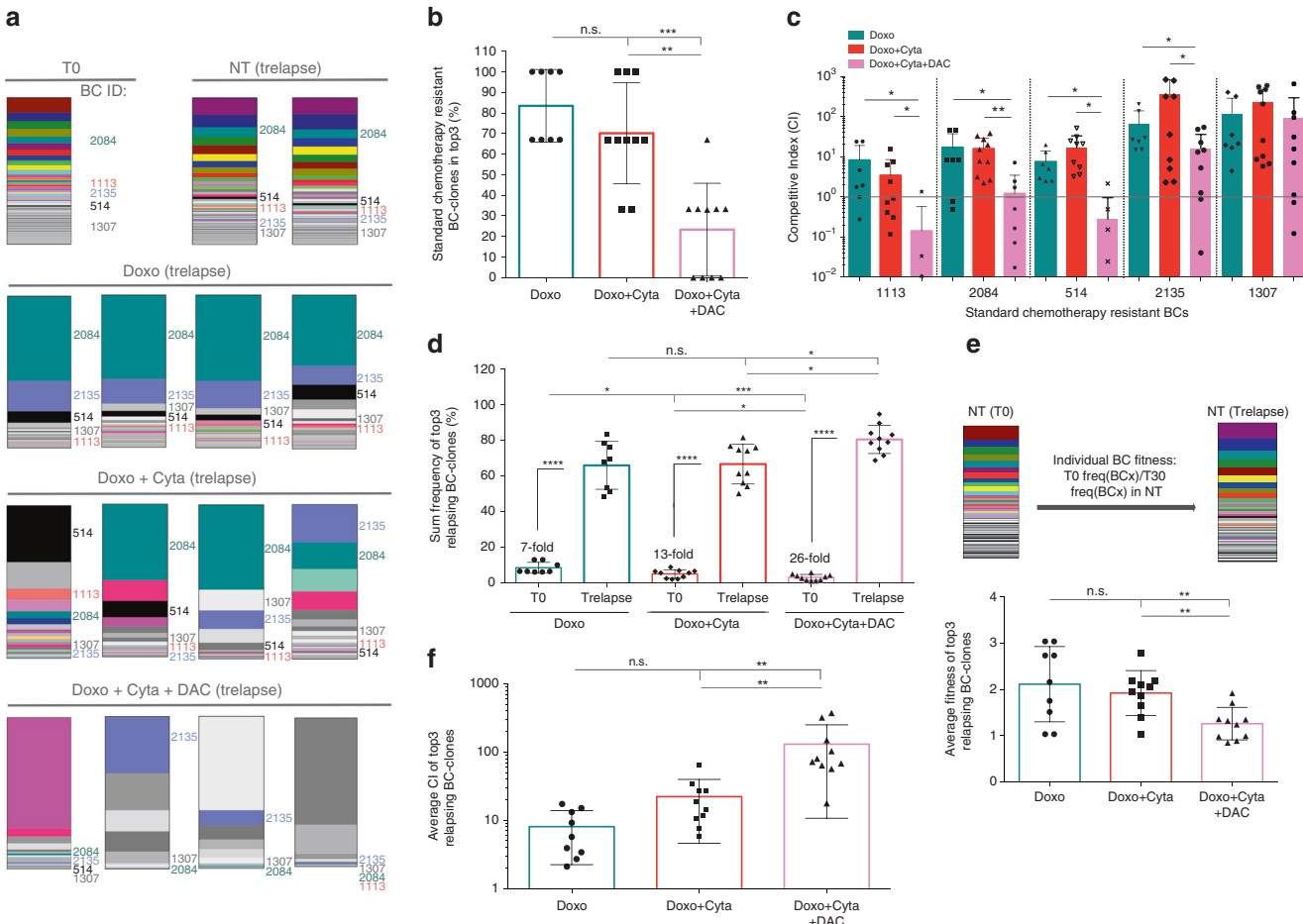

**Fig. 3** Decitabine combination suppresses the re-growth of chemoresistant BC-clones. (Data relative to HEL cell line) **a** BC-clonal composition of representative NT, Doxo, Doxo + Cyta and Doxo + Cyta + DAC samples at Trelapse. Each BC-clone has a fixed color-code across all samples. Chemoresistant BC-clones are indicated. **b** Average frequency at which chemotherapy-resistant BC-clones constitute the top 3 most dominant clones at Trelapse. In total, 3 out of 3 represents 100% frequency, 2 out of 3: 67%, 1 out of 3: 33% and 0 out of 3: 0% (Doxo: $n = 8$, Do33xo + Cyta ± DAC: $n = 10$). **c** Competitive index (CI) of each chemoresistant BC-clone in Doxo ($n = 8$), Doxo + Cyta ($n = 10$) and Doxo + Cyta + DAC ($n = 10$) groups. A CI of 1 (dotted line) represents a similar barcode fold variation form T0 to Trelapse in the indicated condition relative to NT. **d** Summed frequency of top 3 most dominant BC-clones in each treatment group at Trelapse and their matched frequency at T0. The fold variation of the summed frequency of these BC-clones between T0 and Trelapse is indicated. **e** Schematic representation of experimental approach to calculate BC-clone specific fitness. Lower graph—Average fitness of the top 3 most dominant BC-clones from indicated groups (Doxo: $n = 9$, Doxo + Cyta ± DAC: $n = 10$). **f** Competitive index (CI) of the top 3 most dominant BC-clones in indicated groups at Trelapse (Doxo: $n = 9$, Doxo + Cyta ± DAC: $n = 10$). Graphs of mean ± s.d., P values were determined by t-test (panel c.)one-way ANOVA test. ns – not significant, *$P < 0.05$; **$P < 0.01$; ***$P < 0.001$; ****$P < 0.0001$. Source data are provided as a Source Data file

determinants in our model, we performed exome- and RNA-sequencing on Trelapse samples. We observed an overall stability of the genomic composition of NT, Doxo ± Cyta and Doxo + Cyta + DAC samples (Supplementary Fig. 8a–e), corroborated by high correlation between exonic-variant frequencies (in total exome and in AML/cancer gene panels) in the different groups (Supplementary Fig. 8f, g). Despite the overall sharing of exomic variants between all the conditions, we observed exonic variants unique to each condition (Supplementary Fig. 8c), suggesting an active on-going genetic clonal evolution process in our experimental model. However, few of these variants were on genes thought to have a causative role in AML pathogenesis[13], the exception being de novo subclonal *FLT3* mutations of unknown functional consequence and thus likely representing passenger mutations in Doxo + Cyta + DAC relapses (Supplementary Fig. 8f). On the contrary, established AML driver mutations *JAK2*^V617F^ and *P53*^M133K^ (P53 loss of function) were present at variant allele frequencies of 100% in all groups, as expected from their role as founding mutations in HEL cell line[31] we

(Supplementary Fig. 8f). Importantly, there were no common mutations exclusively in chemoresistant groups (Doxo and Doxo + Cyta) in genes known to be mutated in chemoresistance[11]. These observations suggest that factors other than exomic mutation could participate in chemotherapy resistance. To further explore this hypothesis, we preformed transcriptomic analysis on Trelapse samples. Contrary to exomic data, but in agreement with BC-clonal dynamics, RNA sequencing revealed that Doxo + Cyta + DAC samples were clearly more divergent from NT than Doxo ± Cyta groups (Fig. 4a), showing the highest number of statistically significant differentially expressed genes relative to NT (Doxo: 188; Doxo + Cyta: 40; Doxo + Cyta + DAC: 1049—genes with fold change >< than ±3; adjusted p-value < 0.01). Doxo and Doxo + Cyta groups showed a conserved transcriptomic profile, supporting their previously established BC-clonal and functional convergences (Fig. 4a,b). Interestingly, gene set enrichment analysis (GSEA) revealed that Doxo + Cyta + DAC relapses were enriched for cell proliferation pathways (E2F and MYC target genes) and DNA synthesis (Fig. 4b,c). We

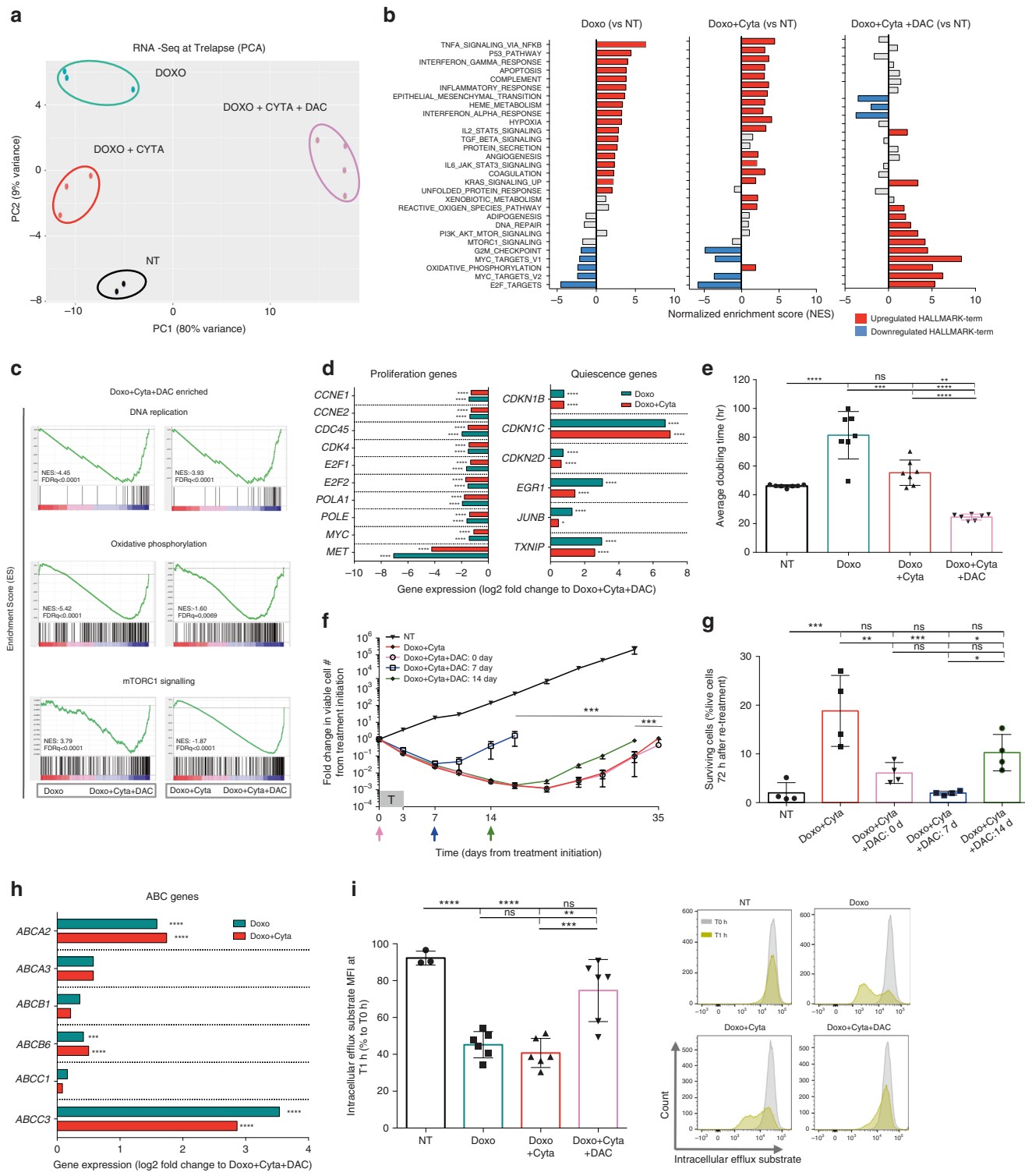

also observed increased expression of pathways related to metabolism such as mTORC1 signaling[32] and oxidative phosphorylation, suggesting an overall increase in proliferative and biogenesis capacity of Doxo + Cyta + DAC relapses compared to both NT and Doxo ± Cyta groups, further reflecting the higher clonal expansion observed in Doxo + Cyta + DAC relapses. In strike contrast, Doxo ± Cyta samples showed not only a decrease in proliferative pathways, but an enrichment in pro-inflammatory pathways together with an up-regulation of the hypoxia pathway (Fig. 4b,c), suggestive of a quiescence state in Doxo ± Cyta relapses[33,34]. This was further supported by the finding that multiple genes involved in cellular quiescence such as *CDKN1b, CDKN2d, EGR-1* or *TXNIP* were enriched in Doxo ± Cyta relapses (Fig. 4d). Strikingly, the most enriched transcript relative to Doxo + Cyta + DAC was *CDKN1C*, a key inhibitor of several G1 cyclin/Cdk and negative regulator of cell proliferation that directly mediates stem-cell quiescence[35,36](Fig. 4d, Supplementary Figure 9a). Conversely, multiple genes involved in cellular

**Fig. 4** Dectabine shapes the transcriptome and proliferative capacity of relapsing hAML. (Data relative to HEL cell line) **a** Principal component analysis plot of differential gene expression at Trelapse between experimental groups. **b** Gene set enrichment analysis (GSEA) of HALLMARK gene set terms significantly (FDRq < 0.05) upregulated (red) and downregulated (blue) in each treatment group as compared to NT, at Trelapse. Gray bars represent no statistically significant change. **c** GSEA of differentially expressed pathways in Doxo + Cyta + DAC group compared to Doxo ± Cyta groups at Trelapse. For GSEA analysis Kolmogorov-Smirnov statistical test was preformed. **d** Differential expression of indicated proliferation and quiescence genes between indicated groups at Trelapse. **e** Doubling time (hours) of indicated groups at Trelapse ($n = 7$). **f** Fold change in total number of viable AML cells between day 0 and 35 after chemotherapy treatment. Arrows indicate time point of DAC (0.1 μM) addition: 0 (pink), 7 (blue), and 14 (green) ($n = 3$). Cells were in all cases exposed to DAC for 72 h. **g** Frequency of viable cells after in vitro (re-)exposure to Doxo + Cyta for 72 h in NT, Doxo + Cyta and Doxo + Cyta + DAC (0, 7 or 14 days) relapsing cells ($n = 4$). **h** Differential expression of indicated ATP-binding cassette transporters (ABC) genes between indicated groups at Trelapse. **i** Intracellular efflux substrate MFI of NT, Doxo ± Cyta and Doxo + Cyta + DAC cells at Trelapse. MFI was determined after 1 h (T1h) at 37 °C and depicted as the ratio of the respective MFIs measured immediately after intracellular efflux substrate staining at T0h ($n = 6$ in all groups except NT: $n = 3$). Representative histograms. Graphs of mean ± s.d. *P* values were determined by one-way ANOVA test. ns—not significant, *$P < 0.05$; **$P < 0.01$; ***$P < 0.001$; ****$P < 0.0001$. Gene expression graphs (**d**, **h**) represent mean log2(fold) change and indicated adjusted *P* value calculated by Wald testing. Source data are provided as a Source Data file

proliferation pathways such as cyclins (*CCNE1,E2*), E2F transcription factors (*E2F1,2*) and DNA polymerase subunits (*POLA1, POLE*) where highly expressed in Doxo + Cyta + DAC relative to Doxo ± Cyta relapses (Fig. 4d). Furthermore, the tyrosine-protein kinase *MET*, a driver of AML proliferation[37] was among the highest expressed transcripts in Doxo + Cyta + DAC relapses. The increased proliferative capacity of Doxo + Cyta + DAC (compared to Doxo ± Cyta) relapses was further substantiated by lower doubling times (Fig. 4e, Supplementary Fig. 9b), increased G2-S-M cell cycle stage frequency (Supplementary Fig. 10a, b), and consistently smaller fractions of undivided cells as assessed by CSFE labeling (Supplementary Fig. 10c). To confirm the ability of DAC to antagonize the quiescent state induced by chemotherapy, we added DAC (0.1 μM) at days 7 or 14 after chemotherapy exposure, and observed markedly earlier hAML cell re-growth compared to chemotherapy alone (Fig. 4f). Critically, whereas Doxo + Cyta relapses acquired chemoresistance, relapses resulting from DAC combination at all tested time points remained as chemosensitive as NT samples (Fig. 4g), suggesting an overt positive association between quiescence disruption and chemosensitivity in our model. Next, to further dissect the mechanisms of chemoresistance we focused the analysis on genes implicated in AML response to chemotherapy. Although there were no differences in the majority of genes associated with doxorubicin or cytarabine resistance[38], we observed that the *ABCG2* gene was significantly up-regulated in all treatment conditions compared to NT (Supplementary Data 1), suggesting that this gene family associates with relapse in our system. Further analysis of the ATP-binding cassette transporters (ABC) gene family revealed that multiple ABC genes implicated in AML relapse and chemoresistance[39] were up-regulated in chemoresistant Doxo ± Cyta groups compared to Doxo + Cyta + DAC, particularly *ABCC3, A2* and *B6* (Fig. 4h). Strikingly this associated with increased transporter activity in the Doxo ± Cyta groups compared to NT and Doxo + Cyta + DAC groups (Fig. 4i), further implicating ABC gene family in chemoresistance development in our system. Overall, these data demonstrate that chemoresistant Doxo ± Cyta relapses display increased quiescence and ABC transporter activity phenotypes that are completely reversed upon DAC combination, leading to highly proliferative chemosensitive relapses. Importantly, these findings also suggest that DAC acts mainly via modulation of the transcriptomic landscape of relapsing hAML cells, rather than impacting on their genetic (mutational) profiles.

**DAC combination reduces stemness properties of hAML relapses.** Low proliferation and quiescent states in leukemic blasts have been mechanistically linked to resistance to cytotoxic drugs[40,41]. Importantly, in AML such low cycling states have been tracked down to sub-populations of leukemic cells that possess

enhanced self-renewal and leukemia-initiating capacity, termed leukemia stem cells (LSCs)[42–44]. In fact, LSC frequency and associated transcriptional signatures carry clinical prognostic impact and have been extensively associated with chemoresistance and leukemia relapse[17–19,45–47]. We thus hypothesized that DAC addition to chemotherapeutic regimens could impact leukemia stemness properties. Consistent with our hypothesis, multiple stem-cell signatures: adult tissue stem-cell[48], hematopoietic[49], mesenchymal[50] and cancer[51,52] stem cell signatures, were all enriched in Doxo ± Cyta compared to Doxo + Cyta + DAC relapses (Fig. 5a). Moreover, the expression of multiple genes associated with hematopoietic and leukemic stem cell self-renewal (*HoxA4,B4,B5; FoxO1,O3,O4; CBX7; ANG; EMCN*) was significantly upregulated in Doxo ± Cyta compared to Doxo + Cyta + DAC relapses (Fig. 5b). Furthermore, Doxo ± Cyta relapses, but not Doxo + Cyta + DAC relapses, displayed a series of stem cell hallmarks: increased expression of LSC markers (% CD34 + CD38− and CD99 MFI; Fig. 5c, Supplementary Fig. 9c), increased ATP-binding cassette (ABC) transporter activity (Fig. 4i) and increased aldehyde dehydrogenase activity (Fig. 5d). A key functional readout of leukemia cell stemness is their ability to initiate leukemia upon transplantation into irradiated immunodeficient mice[53]. To assess the frequency of leukemia-initiating cells (L-ICs) in Trelapse samples, these were engrafted in limiting dilution into sublethaly irradiated NRGS mice (Fig. 5e). Doxo + Cyta + DAC chemosensitive cells showed reduced L-IC frequency when compared to chemoresistant Doxo ± Cyta relapses, leading to less aggressive leukemia development as reflected in increased overall survival of the hosts (Fig. 5f,g). Altogether, our functional assessment of Trelapse cell populations shows that chemoresistance development associates with lower proliferation and increased stemness properties, which are both prevented by up-front combination with DAC. Next, to assess functional leukemia-initiating capacity of individual BC-clones, we performed in vivo establishment of T0 samples with known BC architectures and L-IC frequency (Fig. 5h). By assessing in vivo engraftment of each individual BC in each individual mouse tested, we were able to define a group of 36 BC-clones with significantly higher leukemia-initiating capacity than the population average (HiL-IC BC-clones) (Fig. 5h,i). Quantification of HiL-IC BC-clones present in Trelapse samples confirmed that the majority was efficiently suppressed by DAC combination (Fig. 5j; Supplementary Fig. 9d). Importantly, among the HiL-IC HEL BC-clones, we found four out of five pre-determined chemoresistant BC-clones previously identified in Doxo ± Cyta Trelapses (Fig. 5k; Supplementary Fig. 9e), further demonstrating that chemoresistance strongly associates with in vivo leukemia initiating capacity at both population and BC-clonal levels. Finally, to confirm the ability of DAC combination to target L-ICs, we generated BC-clones from in vivo established single-L-ICs (L-IC

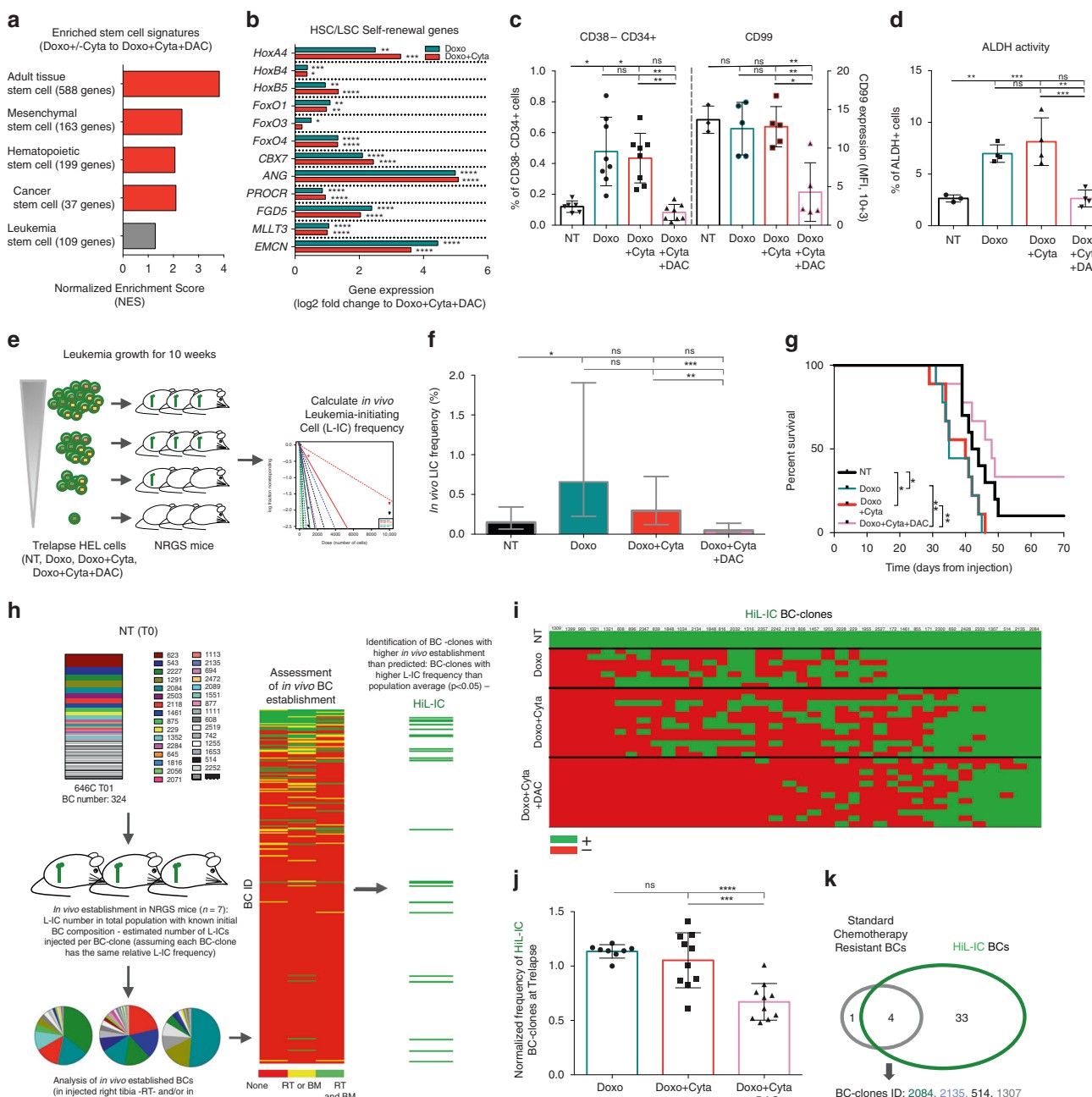

**Fig. 5** Decitabine combination reduces stemness properties of relapsing hAML cells. (Data relative to HEL cell line) **a** GSEA of different stem-cell signatures significantly enriched (red, FDRq < 0.05 calculated by Kolmogorov-Smirnov statistical test) in Doxo ± Cyta Trelapses relative to Doxo + Cyta + DAC group (indicated NES are relative to the highest enrichment observed in Doxo ± Cyta). **b** Differential expression of indicated genes implicated in hematopoietic and leukemic stem cells (HSCs, LSCs) self-renewal between indicated groups at Trelapse. **c** Frequency of CD38-CD34 + cells (left axis, NT: $n = 6$, Doxo ± Cyta and Doxo + Cyta + DAC: $n = 8$) and CD99 mean florescence intensity (MFI) (right axis, NT: $n = 3$, Doxo ± Cyta and Doxo + Cyta + DAC: $n = 5$) on indicated groups at Trelapse. **d** Fraction (%) of cells with active ALDH enzymatic activity in indicated grups (NT: $n = 3$, Doxo ± Cyta and Doxo + Cyta + DAC: $n = 4$). **e** Schematic diagram of limiting-dilution assay performed with Trelapse populations from NT, Doxo ± Cyta and Doxo + Cyta + DAC cells on NRGS mice, via intra-bone marrow injection of 10000, 1000, 100 and 10 cells. **f** Estimated leukemia-initiating cell (LIC) frequency in Trelapse populations from indicated groups (the y-axis denotes the confidence intervals—lower, estimate and upper for L-IC frequency). **g** Survival curves of mice transplanted with 1000 cells from Trelapse NT ($n = 10$), Doxo ± Cyta ($n = 9$) and Doxo + Cyta + DAC ($n = 9$) samples. **h** Schematic diagram of the strategy used to identify BC-clones with a higher L-IC frequency than the population average (HiL-IC). **i** Heat-map scoring the presence (green) or absence (red) of HiLIC BC-clones (columns, total of 37) present in Trelapse populations from replicates of indicated groups (lines). **j** Normalized frequency of HiL-IC BC-clones present at Trelapse in indicated groups (Doxo: $n = 8$, Doxo + Cyta ± DAC: $n = 10$). **k** Venn diagram depicting the overlap across BC-clones with HiL-IC and chemoresistant properties. Graphs of mean ± s.d., $P$ values were determined by one-way ANOVA test. Survival curve analysis was performed by Log-rank (Mantle-Cox) testing. ns—not significant, *$P < 0.05$; **$P < 0.01$; ***$P < 0.001$; ****$P < 0.0001$. Gene expression graph (**b**) represents mean log2(fold) change and indicated adjusted $P$ value calculated by Wald testing. Source data are provided as a Source Data file

BC-clones))(Supplementary Figure 11a,b). In this setting, each BC-clone derives necessarily from an in vivo established L-IC. Strikingly, ex vivo treatment of in vivo established BC cells with Doxo + Cyta + DAC regimen significantly reduced the number of L-IC BC-clones compared to Doxo ± Cyta (Supplementary Fig. 11c), thus confirming the ability of DAC combination to target L-ICs. Our data demonstrates that chemotherapy selects for pre-determined BC-clones with stemness properties, including low proliferation, chemoresistance and leukemia-initiating potential, which are suppressed upon DAC combination that thereby favors the expansion of chemosensitive BC-clones with decreased stemness capacity.

**DAC combination depletes leukemia stem cells in hAML samples.** Having established the capacity of upfront DAC combination to prevent stemness-mediated chemoresistance in our in vitro system, we aimed to validate this finding in a in vivo xenotransplantation model. For this purpose, we tested the effect of standard chemotherapy (Doxo + Cyta) alone or combined with DAC (Doxo + Cyta + DAC) on an in vivo orthotopic model of intra bone-marrow (BM) transplantation of hAML HEL cell line into NRGS mice (Fig. 6a)[54]. Contrarily to the in vitro system where more than 99% of the cells were eliminated by equivalent regimens (Fig. 1b), in vivo cell elimination in the peripheral blood of leukemic animals was mild (Fig. 6b). This resulted mainly from the inability to increase in vivo chemotherapy doses beyond the limit of chemotherapy toxicity in immunodeficient mice[55,56]. Notwithstanding this limitation, mice treated with Doxo + Cyta + DAC showed increased leukemia cell elimination and overall survival compared to untreated (NT), DAC alone and Doxo + Cyta treated mice (Fig. 6b,c). Next, to assess cell-intrinsic gain of chemoresistance, we sorted BM-resident hAML cells from non-treated (NT) and Doxo + Cyta ± DAC treated mice and re-exposed them to chemotherapy (Doxo + Cyta) ex vivo. Strikingly, whereas in vivo chemo-exposed AML cells showed a three-fold gain of resistance, co-exposure to DAC preserved NT-like sensitivity upon re-treatment (Fig. 6d). Despite the lower magnitude of chemoresistant gain in vivo compared to the in vitro system (likely resulting from the limited amount of chemotherapy used), these data confirmed the overall capacity of upfront DAC combination to prevent chemoresistance development in hAML cells. Importantly, by assessing the frequency of immunophenotypically defined LSCs (live CD38-CD34 + cells) in NT, Doxo + Cyta and Doxo + Cyta + DAC, we could confirm that DAC combination prevents the chemotherapy-induced increase in LSC frequency (Fig. 6e). Altogether, these data validate our in vitro observations on the effect of DAC combination on hAML stemness and chemoresistance at relapse in a more physiological in vivo xenotransplantation model. Finally, we tested the effect of DAC combination in targeting immunophenotypically defined LSCs (live CD33 + CD38- CD34 + cells) from human primary AML samples (Supplementary Figure 12; Supplementary table 1). Bone marrow AML blasts were cultured ex vivo in the presence of no treatment (NT), Doxo + Cyta or Doxo + Cyta + DAC for 3 consecutive days; and 3 days after drug withdrawal the absolute number of viable CD33 + CD38- CD34 + LSCs was assessed (Fig. 6f). We observed that while Doxo + Cyta depleted LSCs numbers in the majority of the samples, the combination with DAC clearly enhanced this effect, leading to a significant decrease in LSC numbers in 6 out 8 tested samples (samples 1, 3, 12, 15, 20, and 25; Fig. 6g,h). These data further strengthen our observations that upfront combination of low-dose decitabine with chemotherapy depletes LCS numbers in hAML relapses, thus attesting the potential of this approach to tackle stemness-associated chemoresistance development in AML.

## Discussion

Therapeutic resistance drives recurrences and represents a major hurdle to successful clinical management of cancer in general, and AML in particular. In this study we have established an experimental model system to characterize longitudinally the clonal dynamics and associated genetic and non-genetic determinants underlying chemoresistance development in hAML cells exposed to different chemotherapeutic regimens. Using in vitro lineage tracing coupled with exome, transcriptome and in vivo functional readouts we revealed the ability of low-dose DNMTis in upfront combination with chemotherapy to prevent chemoresistance development by suppressing the expansion of a pre-determined set of AML clones with high stemness properties. These data represent a major advance to our understanding of the clinical benefits described for such combinatorial regimens in refractory/relapsed AML[25–27].

The use of DNMTi to sensitize chemoresistant tumors has been applied in different cancer types with some success, having been associated with transcriptional activation of tumor suppressor genes such as *TP53*[57]. Here, we found that combining chemotherapy with DAC leads to major transcriptomic and functional changes of relapsing hAML cells, leading to an overall decrease in stemness properties (quiescence, drug efflux capacity and leukemia-initiating/self-renewal capacity), which have been linked with poor clinical outcomes and chemoresistant relapse development in AML[58]. Among these properties, low proliferation and quiescent states in leukemic blasts have been directly associated with resistance to cytotoxic drugs, as chemotherapeutic agents preferentially target dividing cells[40,41]. Supporting this, we observed that concomitantly with multiple regulators of cellular quiescence, the *CDKN1C* gene, a key negative regulator of cell proliferation that directly mediates stem-cell quiescence and self-renewal capacity in hematopoietic stem cells (HSCs)[35,36], was the most enriched transcript in chemotherapy relapses when compared to DAC combination samples. High CDKN1C expression levels in BM associate with lower proliferative activity and poor survival after standard chemotherapy in both AML and MDS patients[59]. This further suggests that reduced proliferation in chemotherapy-driven relapses may represent a mechanistic pathway of resistance that is prevented by DAC addition, thus resulting in proliferative and chemosensitive relapses. Importantly DAC combination also associated with an overall decrease in the expression of multiple ABC gene family members (in particular *ABCC3*) which are well established regulators of chemoresistance in AML[39,60]. Another key transcriptional and functional aspect of chemoresistant hAML relapses in our model was increased leukemia-initiating capacity (in immunocompromised mice) and self-renewal gene expression, which was strikingly abolished upon DAC combination. This observation is in line with previous studies showing that DNA methylation promotes self-renewal and inhibits differentiation of both HSCs and LSCs;[61,62] and that low-dose DNMT inhibitors can reduce the tumorigenicity of cancer stem cells in various models[63]. The molecular mechanism by which DNMTi combination impacted on the expression of stemness-associated genes was not established in our study. A potential explanation for the observed increase in proliferation upon DNMTi combination is the hypomethylation-dependent enhanced transcription of pro-cycling genes (e.g. *MET*[64]), whereas repressed signature genes (e.g. *CDKN1C, ABCC3, CBX7*) may be indirectly regulated via hypomethylation-promoted transcription of their negative regulators. Critically, our study reveals for the first time the capacity of DAC in upfront combination with chemotherapy to reduce the stemness properties of hAML relapses, further encouraging the clinical assessment of upfront combination of low-dose DNMT inhibitors with standard chemotherapy as first line treatment of AML.

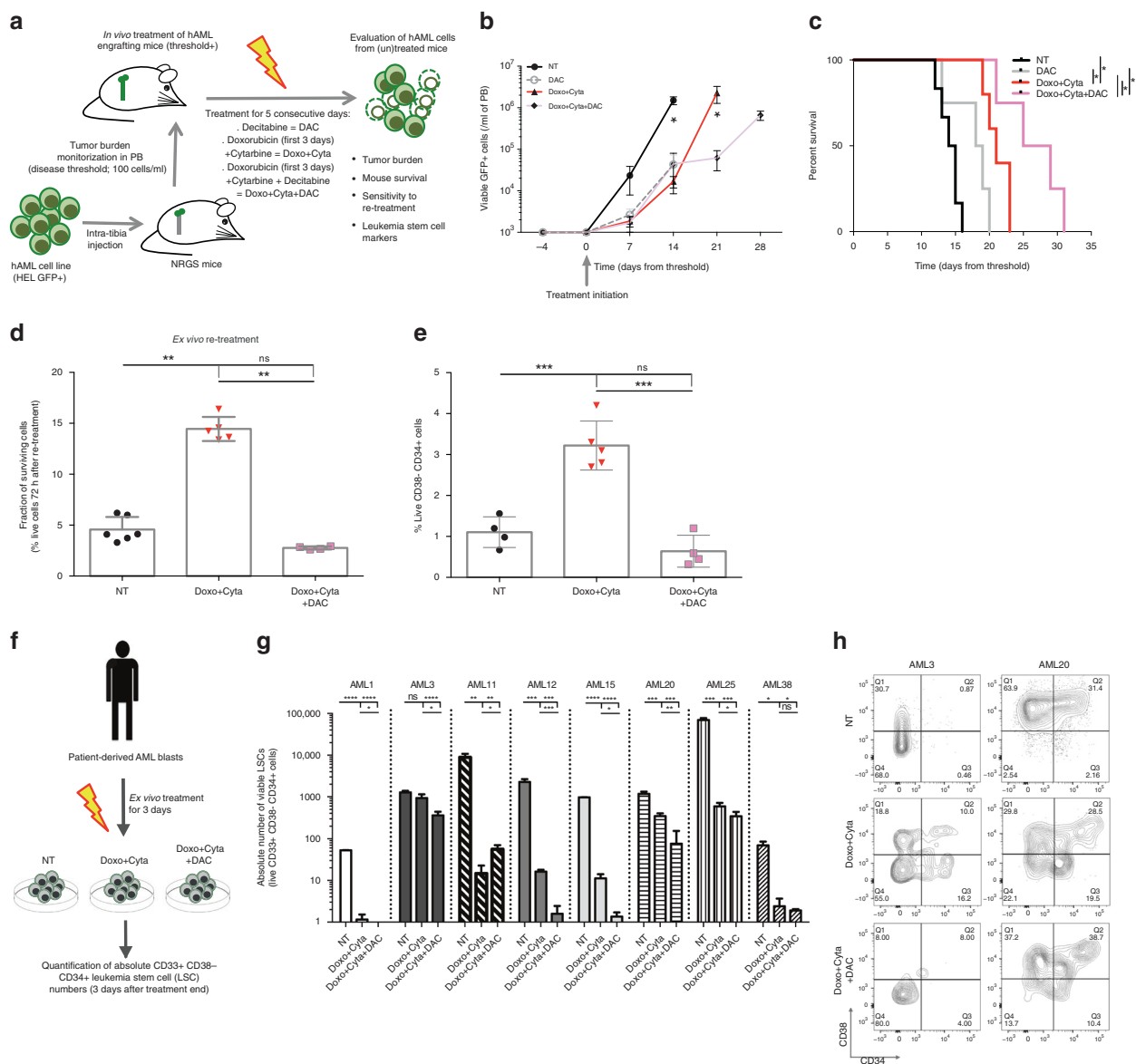

**Fig. 6** Decitabine combination depletes LSCs from xenografts and AML patient samples. **a** Schematic diagram of the therapeutic setting used to treat NRGS mice bearing human AML (GFP + HEL) cells. Chemotherapy regimen (Doxo + Cyta): intraperitoneal (i.p.) injection of cytarabine (100 mg/kg per day over 5 days) and doxorubicin (3 mg/kg per day over the first 3days). Decitabine combination was initiated 2 days after Doxo + Cyta, via i.p. injection of 0,5 mg/kg DAC for 5 consecutive days. **b** The total number of viable hAML HEL (GFP + , 7AAD-) cells in peripheral blood (PB) and mouse survival (**c**) in untreated (NT, n = 6), DAC alone (n = 4), Doxo + Cyta (n = 5) and Doxo + Cyta + DAC (n = 4) groups. **d** Frequency of viable BM-sorted hAML cells, sorted from the BM of mice from the three groups described in **b**, **c**, after ex vivo exposure to Doxo + Cyta for 72 h. **e** Frequency of CD38-CD34 + cells (left axis, n = 4–8) on BM-sorted hAML cells from indicated groups. **f** Schematic diagram depicting ex vivo treatment of AML patient samples Doxo + Cyta and Doxo + Cyta + DAC and the subsequent quantification of immunophenotypically defined LSCs. **g** Absolute number of LSCs (live CD33 + CD38− CD34 + cells) after ex vivo exposure to Doxo + Cyta and Doxo + Cyta + DAC (n = 3 per sample). **h** Representative flow cytometry contour plots of CD38/CD34 staining gated on CD33 + VioletZombie- population. Graphs of mean ± s.d., P values were determined by one-way ANOVA test. Survival curve analysis was performed by Log-rank (Mantle-Cox) testing. ns—not significant, *P < 0.05; **P < 0.01; ***P < 0.001; ****P < 0.0001. Source data are provided as a Source Data file

To resolve the underlying clonal dynamics associated with the selection of chemoresistance we employed an in vitro system using barcoded hAML cells. In spite of its inherent limitation in modeling the complex biology of AML cells (e.g. bone marrow niche interactions[65]) this experimental setup offers two key advantages compared to in vivo systems: (1) drug doses are not limited by the sensitivity of immunocompromised mice to cytotoxic agents[56]; (2) stable steady-state clonal dynamics, which is lost upon in vivo xenotransplantation of human cancer cells into immunocompromised mouse models[66] (our unpublished data), allowing for reproducible clonal dynamics assessment upon

experimental perturbations. The latter is particularly relevant since, by creating replicates with a constant barcode composition, it allows testing the effect of different therapies on the same complex tumor population that would not be possible in the clinical setting. Using this unique feature of our system, we found a pre-determined set of BC-clones that is consistently present in chemotherapy relapsing populations, indicating that, even in the absence of selective pressure, there are clonally related cells (BC-clone) that share the molecular determinants that will lead to their selection upon chemotherapy exposure. Interestingly, molecular characterization of the mutational spectrum by exome

sequencing showed a lack of genetic drivers for chemoresistance and an overall conserved exomic landscape in chemosensitive and chemoresistant groups. In stark contrast, transcriptomic assessment revealed a very dynamic shift in gene expression with a strong enrichment in stemness-associated pathways in chemoresistance hAML cells. Importantly, we observed that this pre-determined set of chemoresistant BC-clones also displayed increased capacity to establish in in vivo immunocompromised mice (a classic LSC feature) prior to chemotherapy, indicating that transcriptional/functional defined stemness and chemoresistant states pre-exist in hAML BC-clones and are actively selected by chemotherapy. This indicates that hAML chemoresistance development pre-exists therapy exposure and results from transcriptomic rather than genetic selection, in line with recent studies implicating epigenetic reprogramming of transcriptomic heterogeneity at single-cell level in the development of therapeutic resistance, independently of genetic alterations[67]. Strikingly, our data revealed that upfront DNMTi combination impaired the selection of the chemoresistant pre-determined set of BC-clones, and dramatically favored the expansion of a rarer set of clones that would be normally (i.e., in absence of this specific pressure) outcompeted. This was accompanied by a strong transcriptomic signature of proliferation pathways, further substantiating the clonal expansion observed in these populations upon relapse. These data strongly indicate that DAC uniquely affects the re-growth capacity of each individual BC-clone in a differential manner, causing a significant impairment on the re-growth capacity of pre-determined BC-clones associated with chemoresistance, while favoring the expansion of rarer BC-clones that remain chemosensitive. Although the mechanistic basis of this differential impact of DAC on different BC-clones was not completely clarified here, it is very likely to depend on clonal-specific epigenetic states[22]. In fact, our data suggests a model where, in response to chemotherapy, hAML regrowth is heavily dependent on, and potentially driven by, LSCs—as shown by a clear enrichment of stemness at relapse. Given the high sensitivity of CSCs to DNMTis[61–63] it is therefore possible that in the presence of DNMTi relapses no longer derive from LSCs, but are now largely driven by cells with lower stemness that regenerate the leukemia via activation of proliferative pathways while remaining sensitive to chemotherapy.

In summary, our study demonstrates the ability of low-dose DNMTi in combination with chemotherapy to dramatically shape AML clonal dynamics, markedly depleting a pre-determined set of chemoresistant clones with increased leukemia-initiating potential, thus leading to chemosensitive relapses. Importantly, we show that these clonal dynamics are not based on genetic differences, but instead on the transcriptional landscape associated with stemness and chemoresistance. We thus provide mechanistic insight for the promise of upfront addition of low-dose DNMTi to standard chemotherapy, to sensitize AML to (re)treatment; and propose this might circumvent development of chemoresistance in AML relapses, potentially turning this fatal malignancy into a chronic manageable disease.

## Methods

**Ethics statement**. Bone marrow samples of adult AML patients were collected at the Hematology Department at Instituto Português de Oncologia (IPO, Lisbon, Portugal) after written and informed consent and ethical Review Board approval from Instituto Português de Oncologia, in accordance with the Declaration of Helsinki. Samples used in this study were selected for high frequency of immunophenotypically leukemic blasts (over 80%). All animal experiments were conducted in accordance with standard institutional animal care procedures and followed ethical committee protocols at Instituto de Medicina Molecular João Lobo Antunes.

**Barcode construction and lentivial-barcode production**. The DNA Barcode library containing ~2600 unique barcodes was produced as previously described[68].

To produce barcode containing viral particles HEK 293 T (ATCC® CRL-3216™) cell line was used. Cells were plated at 70.000 cells/cm² in tissue culture flask in IMDM complete medium composed by Iscove's Modified Dulbecco's Medium GlutaMAX™ Supplement (Gibco®, NY, USA) supplemented [10% (v/v) fetal bovine serum (Gibco®, NY, USA), 1% (v/v) Sodium Pyruvate (Gibco®, NY, USA), 1% (v/v) penincillin-streptomycin (Gibco®, NY, USA) and 0.1% (v/v) Gentamycin (Gibco®, NY, USA)] during 16–24 h before transfection, at 37 °C and 5% $CO_2$. For cells transfection 0.0272 μg/cm² of Barcode library expression vector, 0.0181 μg/cm² of gag-pol expressing vector and 0.0181 μg/cm² of VSVG expressing vector were mixed with 6.53 μL/cm² serum free medium and 1.11 μL/cm² Lipofectamine 2000 (InvitrogenTM, NY, USA) and incubated for 20 min at room temperature (RT). Subsequently the medium was removed from the cells and the mixture was immediately added at 14.2 μg/cm² drop-wise and the cultures incubated at 37 °C and 5% $CO_2$ for 24 h. Old medium was removed and 0.2 mL/cm² of IMDM complete medium supplemented with 10 mM Sodium Butyrate (Sigma®, Darmstadt, Germany) was added to the cells and incubated at 37 °C and 5% $CO_2$ for 8 h. Cells were washed 2-3x and 0.133 mL of complete medium/cm² was added. After 24 h of incubation, medium contained viral particles was harvested and filtrated with 0.45 mm filter, aliquoted and frozen at −80 °C.

**Barcoding of AML cell lines**. Human acute myeloid leukemia cell lines HEL (ATCC, TIB-180™) and OCI-AML3 (DSMZ, ACC 582) were maintained in Iscove's Modified Dulbecco's Medium GlutaMAX™ Supplement supplemented with 10% (v/v) fetal bovine serum, 1% (v/v) Sodium Pyruvate (Gibco®, NY, USA), 1% (v/v) penincillin-streptomycin and 0.1% (v/v) Gentamycin (Gibco®, NY, USA) at 37 °C with 5% $CO_2$. Cells were seeded at $0.2 \times 10^6$ cells/mL and maintained between $0.2 \times 10^6$ cells/mL and $0.6 \times 10^6$ cells/mL. To transduce the AML cell lines, $6 \times 10^6$ cells were plated per well in a 24-well plate with IMDM complete medium containing 12 μg/μL of polybrene (Merck®, Darmstadt, Germany), then the supernatant of viral particles previously obtained was also supplemented with 12 μg/μl of polybrene and added to the cells. Control wells without viral supernatant were prepared in parallel. The plates were centrifuged at 1500 g for 90 min at 32 °C. Finally, the supernatant was removed, the cells were washed twice by changing the medium and centrifugation at 350 g for 7 min. The cells were incubated at 37 °C and 5% $CO_2$ for 20 h. Initially, serial dilutions of the supernatant were tested, starting in a 1:2 dilution in order to determine the amount of virus that will transduced a maximum of 5% of the cells in the first 24 h. The <5% level of transduction was previously determined to be the level at which >1 barcodes are very rarely integrated per cell. In detail, we preformed double transductions using GFP + and RFP + coding viral vectors and determined that double transductions (as measured by dual positivity) were only observed above ~10% levels of co-transduction. After 20 h incubation transduced cells were collected, washed, stained with 10 μl/mL of 7AAD (Biolegend, USA) and 2.5 μL/mL of AnnexinV-APC (InvitrogenTM, NY, USA) and sorted in a FACSAriaTM III BD (Biosciences). Viable cells were sorted and seeded at different numbers in independent wells. After cell expansion the barcode number of each independent culture was determined and only sublines with approximately 10% of the barcodes in the library (210–330 barcodes) were used as T0 samples. This level of library usage was previously determined to prevent that the same barcode is found in 2 independent cells transduced with the same library[58]. Multiple aliquots of T0 samples were stored at −80 °C in fetal bovine serum supplemented with 10% DMSO (Sigma®, Darmstadt, Germany) to be used as T0 for all experiments. HEL, HEK 293 T and OCI-AML3 were purchased from ATCC and DSMZ respectively (cell lines were not tested for mycoplasma, nor authenticated).

**Xenotransplantation of barcoded AML cells**. NOD-Rag1-/-γc-/- (NRGS, 024099) mice were obtained from the Jackson Laboratories (USA). 8–10-weeks-old male and female NRGS mice were sub-lethally irradiated (250 cGy) 24 h before the injections and kept with oral antibiotic (Bactrim) diluted in the drinking water during all the experiment. For the xenotransplantation, a previously described model of intra bone marrow (IBM) xenografted human AML cells was used[54]. Briefly AML cells (barcoded HEL and OCI-AML3 cell lines) were washed, resuspended in PBS and injected directly into the right tibia bone marrow 10000 cells/mouse. Mice were housed at IMM Lisboa, monitored daily for signs of disease and weekly for tumour burden levels quantification. Animals exhibiting signs of disease (paralysis, more that 20 % of weight loss or high tumour burden) were killed by $CO_2$ and the bone marrow and spleen were immediately collected for posterior analysis.

**In vivo leukemia progression evaluation**. Leukemia progression was evaluated by weekly quantification of peripheral blood circulating tumour cells. Blood (70–100 μL) was collected from facial vein and stored in tubes containing 10 μL of Heparin (500un/mL, B.Braun, Germany). In each blood sample, a defined amount of beads (Coulter CC Size Standard L10) was added for absolute cell quantification and Red Cells Lysis Buffer 1x(RBC Lysis Buffer 10x, Biolegend, USA) was added in multiple rounds until complete erythrocyte clearance. Cells were stained with LIVE/DEADTM dead cell staining kit (InvitrogenTM, NY, USA) for 15 min at 4 °C. Total GFP + Live/Dead– cells were quantified by Flow cytometry on an LSR FortessaII Cell Analyzer (BD Biosciences). Data was analyzed with FlowJo X 10.0.7 software (TreeStar, USA) and the results shown as the absolute numbers per mL of blood.

Animals were assigned randomly to treatment groups upon reaching a minimum level of 100 GFP+ CD45+ viable cells/mL of blood (treatment threshold).

**In vivo chemotherapy drugs treatment**. NRGS mice were IBM xenotransplanted with 10000 cells before treatment assignment. Upon reaching tumor load threshold levels mice were treated with: (1) chemotherapy regimen (Doxo + Cyta): intraperitoneal (i.p.) injection of cytarabine (100 mg/kg per day over 5 days—Citaloxan 20 mg/kg stock) and doxorubicin (3 mg/kg per day over the first 3days—doxorubicin chlorohydrate, 2 mg/mL stock) as previously described[55]. (2) Chemotherapy combined with decitabine (Doxo + Cyta + DAC): chemotherapy was administred as in 1) and decitabine combination was initiated 2 days after Doxo + Cyta, via i.p. injection of 0.5 mg/kg DAC (5-Aza-2′-deoxycitidine, 5 mg, Sigma) for 5 consecutive days[69]. (3) No treatment group (NT) received i.p. PBS. Disease progression was monitored as described. Mice were sacrificed upon signs of illness and viable GFP + AML cells were bone marrow sorted and stored at −80 °C, for further ex vivo evaluation.

**In vivo leukemia-initiating cells evaluation**. NRGS mice were xenotransplanted with serial dilutions (100000, 10000, 1000, 100, 10 cells) of barcoded HEL and OCIAML3. Starting populations were in vitro generated NT, Doxorubicin (Doxo), Doxorubicin plus Cytarabine (Doxo + Cyta) or Doxorubicin plus Cytarabine and Decitabine (Doxo + Cyta + DAC) relapsing cells. Mice were sacrificed upon signs of illness or upon experiment termination (10 weeks after xenotransplantation). Viable GFP + AML cells were quantified in the injected tibia or in the pool of the remaining tibia, left and right femurs. Animals with human AML cells frequency over 0.5% of the total mouse leukocyte population in either injected tibia or pooled bone marrow were scored as positive. Leukemia-initiating cell (L-IC) frequency was determined by Extreme Limiting Dilution Analysis (ELDA) software (http://bioinf.wehi.edu.au/software/elda/index.html), provided by the Walter and Eliza Hall Institute[70].

**Determination of clones with leukemia-initiating potential**. NRGS mice were xenotransplanted with 10000 barcoded HEL (total of seven mice) or OCI-AML3 (total of six mice) NT cell populations with known barcode (BC) architecture and predicted average number of L-ICs. An estimated number of L-ICs per BC-lineage present in the 10,000 xenotransplanted cell population was determined assuming even distribution of L-ICs by the different BC-lineages. After determining the BC arquitecture of in vivo established NT HEL and OCI-AML3 cell populations a Poisson-probability distribution criteria was defined to establish which BC-lineages were more frequently found in vivo (i.e. established in more mice) than expected. These lineages were defined as high leukemia-initiating cell (HiL-IC) lineages.

**In vitro chemotherapy treatment of hAML cell lines**. All the in vitro treatment regimens were optimized using the mainstream drugs used in clinical management of AML. The different treatment regimens consisted in using only the anthracycline doxorubicin (1.8 μM, doxorubicin chlorohydrate, 2 mg/mL, medac), cytarabine (6 μM, Citaloxan, 20 mg/mL, Hospira Portugal Lda) or hypomethylating agent decitabine (0.1 μM, 5-Aza-2′-deoxycytidine, 5 mg, Sigma) and using conjugations of this drugs with or without the hypomethylating agent (namely, doxorubicin plus cytarabine and doxorubicin plus cytarabine combined with decitabine). Non-treated cells were cultured in IMDM with 0.1 μM of dimethyl sulfoxide (DMSO), as control. The drugs were always added to the cells at time T0h (beginning of each experiment) and each treatment condition was always performed at least in three biological replicates per experiment. Barcoded HEL cells were counted by Trypan blue exclusion method (>75 % of viability was ensured), and 6 million cells were seeded at the concentration of $2 \times 10^5$cells/mL in the presence or absence of each treatment regimen and incubated for 72–74 h at 37 °C, 5% CO$_2$. After incubation, cells were thoroughly washed and enriched for live cells through Ficoll-Paque (Histopaque®-1077, sigma) gradient exclusion of dead cells. The recovered live cells were washed three times and reseeded at the concentration of $2 \times 10^5$ cells/mL in fresh IMDM complete medium. Culture cells were kept growing at 37 °C, 5% CO$_2$ during approximately 30 days until the number of live cells reached the initially seeded number (6 million cells). Every 3 or 4 days 1% of each cell culture was stained with 10 μL/mL of 7AAD (Biolegend, USA) and 2.5 μL/mL of AnnexinV-APC (InvitrogenTM, NY, USA) and analyzed for the total cell numbers and the live/dead cells by flow cytometry on an LSR FortessaII Cell Analyzer (BD Biosciences). Pellets of live cells ($1 \times 10^4$-$1 \times 10^5$) were collected and frozen for barcode sequencing at indicated timepoints. Cultures that regrew from previous treatment regimens were (re)treated only with chemotherapy for 72 h.

**RNA isolation, cDNA production, and real-time PCR**. mRNA was extracted from cell lines using High Pure RNA Isolation kit (Roche). Reverse transcription was performed with random oligonucleotides (Invitrogen) using Moloney murine leukemia virus reverse transcriptase (Promega) for 1 h at 42 °C. Relative quantification of specific cDNA species to endogenous reference human GAPDH was carried out using SYBR on ABI ViiA7 cycler (Applied Biosystems). The CT for the target gene was subtracted from the CT for endogenous references, and the relative amount was calculated as 2−ΔCT. Primer sequences were the following: GAPDH forward, CTCCTCTGACTTCAACAGCGACAC, GAPDH reverse, TGCTGTAGCCAAATTCGTTGTCAT, CDKN1C forward, AGAGATCAGCGCC TGAGAAG, reverse, GGGCTCTTTGGGCTCTAAAC.

**Analysis of stem cell markers and efflux activity**. Untreated cells (NT) and treatment-relapsing cells (doxorubicin, cytarabine, doxorubicin plus cytarabine and doxorubicin puls cytarabine combined with decitabine) from barcoded HEL and OCIAML3 cell lines were analysed. Stem-cell surface markers were evaluated by flow cytometry using anti-hCD34-APC (Biolegend, USA), anti-hCD38-PE (Biolegend, USA) and anti-hCD99-PE (Biolegend, USA) antibodies. ABC transporter activity was analysed using the eFluxx-ID Gold multidrug resistance kit (Enzo Life Sciences) according to the manufacturer's instructions.

**Quantification and statistical analysis**. All P values were calculated using one-way ANOVA test with GraphPad Prism software, unless otherwise described in the methods or figure legends. No specific randomization or blinding protocol was used for these analyses. Statistically significant differences are indicated with asterisks in figures with the accompanying P values in the legend. Error bars in figures indicate SD for the number of replicates, as indicated in the figure legend.

**Cytosine DNA methylation quantification**. Dry pellets of $0.5–1 \times 10^5$ live cells were prepared for every condition tested. Cell pellets were treated with RNaseA for 1 h at 37 °C and proteinase K for over-night. Genomic DNA was isolated using phenol chloroform and resuspended in 0.01 M Tris-HCL (pH8) and subsequently quantified using dsDNA BR Assay Kit (Qubit). 1 μg of genomic DNA was used for sample preparation by treating with 5U DNA Degradase Plus at 37 °C for 1 h (ZymoResearch, E2021) to obtain individual nucleosides in a final volume of 25 μL and inactivated by adding 175 μL of 0.1% formic acid[71]. DNA-me measurements were performed at the VBCF- Vienna Biocenter Core Facilities.

**Barcode PCR amplification, deep sequencing and analysis**. Barcode quantification was performed as previously described[72]. In detail, dry pellets of $0.5–1 \times 10^5$ live cells were resuspended in 40 μL of DirectPCR® lysis buffer (Viagen Biotech, USA) containing 200 mg/mL of proteinase K. The cells were lysed in a thermocycler at 55 °C for 1 h and 90 °C for 30 min. For the first PCR, 3 μL of TopLib 5′-TGCTGCCGTCAACTAGAACA-3′ and 3 μL of BotLib 5′-GATCTCGAATCAGGC GCTTA-3′ primers and 50 μL of 2x MyTaqTM red Mix (Bioline, UK) were added to the 40 μL pellet of each samples. After mixing and before PCR run, 50 μl was transferred to an empty PCR tube, to provide technical replicates. The tubes were placed in a thermocycler initially for 5 min at 94 °C, then 30 cycles of 58 °C for 15 s, 72 °C for 15 s and 94 °C for 15 s were performed and finally for 10 min at 72 °C. For the second PCR different index primers were used for every sample and technical replicate. The index forward primers was designed taking into account a production of a library of 384 different 82-bp primers containing unique 8-bp sequence that differed by at least 2 bases, a P7 annealing region for the Illumina® Sequencing system (Illumina® Sequencing, USA)), and a 16-bp annealing region to the first PCR. For the common reverse primer, a sequence that includes a P5 annealing region for the Illumina® Sequencing system followed by an annealing region for the first PCR product 5′-CAAGCAGAAGACGGCATACGAGATTGCTGCCGTCAACTAGAACA-3′ was designed. For the PCR mixture, 1 μL of the first PCR product, 5 μL of index primer, 1 μL of common primer, 15 μL of 2x MyTaqTM red Mix (Bioline, UK) and 8 μL of H2O were mixed together. PCR amplification was carried out as described above, and the presence of the expected 224-bp product was checked for each sample by 2% agarose gel electrophoresis. After the two PCR, 4 μL of each sample containing different indexes up to 384 that were pooled, run on an E-Gel® Size Select 2% (InvitrogenTM, NY, USA) to obtain the 224-bp product and sequenced on a HiSeqTM 2000 (Illumina® Sequencing, USA). ASCIDEA Computational Biology Solutions, Barcelona, performed the sequencing. A single-run 50-bp sequencing run was sufficient to read through the index, common annealing region, and the first 15 bp of the barcode required for data analysis. Upon sequencing of the barcodes present in each subculture, we analyzed the resulting raw data with a bioinformatic pipeline previously optimized. Barcode sequences were extracted using XCALIBR program (developed at Netherlands Cancer Institute - Genomics Core Facilty) generating tabulated data into a matrix containing the fraction of reads for each barcode versus the indexes. Data was further processed using a customized script in R (kindly provided by Dr. Leïla Prerie, Institute Curie) using three main steps. First excludes samples where there were insufficient read counts from the deep sequencing (average of the two technical replicates <$1 \times 10^4$). Samples having passed this step were then normalized to $1 \times 10^5$ for each sample. Secondly, as a measure of sufficient recovery for subsequent lineage comparisons, we further excluded samples where the two technical replicates did not pass a Pearson correlation coefficient of 0.8. After, all reads of barcodes present in only one of either technical replicate of a given sample—an indication that there was a low confidence for inclusion of that barcode—were changed to zero (0) reads for that sample and excluded. After confirming that technical replicates were well represented their average was then taken for further analysis.

**Total RNA extraction, sequencing, and analysis**. Total RNA was extracted from frozen cells pellets using the High Pure RNA Isolation Kit (Roche, Basileia, Swiss) and quantified using QubitTM RNA HS Assay Kit (InvitrogenTM, NY, USA) on the Qubit® 2.0 Fluorometer (InvitrogenTM, NY, USA). ASCIDEA Computational Biology Solutions, Barcelona, performed the sequencing. One microgram of

high-purity total RNA (defined as having an RNA integrity number greater than 7.0) was used as input for the Illumina TruSeq RNA Sample Prep Kit, Sets A/B (48Rxn) (Illumina). The gel-free protocol was employed for the TruSeq RNA Sample Prep Kit per the manufacturer's specifications and performed on the Beckman Coulter Biomek FXp robotics platform. The standard RNA-fragmentation profile was used as recommended by Illumina (94 °C for 8 min). The PCR-amplified RNA-seq library products were then quantified using the Fragment Analyzer Standard Sensitivity NGS Fragment Analysis Kit (Advanced Analytical Technologies). The samples were diluted to 10 nM in EB Buffer (Qiagen), denatured, and loaded at 2.75 pM on an Illumina HiSeq2000 in Rapid Run Mode using TruSeq Rapid PE Cluster Kit–HS and TruSeq Rapid SBS Kit–HS (200 cycle) reagents (Illumina). The RNA-seq libraries were sequenced at 100 bp paired-end with a 7-bp index using the standard Illumina primers. Quality of the reads obtained by HiSeq2000 sequencing was checked with FastQC software (http://www.bioinformatics.bbsrc.ac.uk/projects/fastqc/). Preprocessing of the reads was performed with fastx-toolkit (http://hannonlab.cshl.edu/fastx_toolkit/index.htmL) and aScidea specific perl scripts property of aScidea (http://www.ascidea.com) in order to filter regions of low quality. Adaptors and low quality bases at the ends of sequences and reads with undetermined bases or with 80% of their bases with less than 20% quality score were trimmed. Raw reads that passed the quality filter threshold were mapped using Bowtie2 2.2.8 to generate read alignments for each sample. The reference genome used was the *Homo Sapiens* version GRCh38. The transcript isoform level and gene level counts were calculated using FeatureCounts from SubRead Package. Differential transcript expression was then computed using DESeq2. The resulting lists of differentially expressed isoforms were filtered by ln(fold_change) > 1 and < −1 and adjusted *p*-value of 0.05. Gene set enrichment analysis (GSEA) was performed with the GSEA v2.0 software (Broad Institute of MIT (Massachusetts Institute of Technology) and Harvard, http://www.broad.mit.edu/gsea) on pre-ranked lists of differentially expressed genes. Normalized enrichment scores (NES) with *P* values < 0.05 and false discovery rates (FDR) < 0.05 were considered statistically significant. Initial GSEA analysis was performed on HALLMARK gene set terms[73].

**Exome sequencing and analysis**. Total DNA was recovered by DNeasy Blood & Tissue Kit (Qiagen, USA) and quantified using QubitTM dsDNA HS Assay Kit (InvitrogenTM, NY, USA) on the Qubit® 2.0 Fluorometer (InvitrogenTM, NY, USA). Fragmentation of 1 μg of genomic DNA was performed using adaptive focused acoustic technology (AFA; Covaris). The fragmented DNA was repaired; an"A"is ligated to the 3′ end, agilent adapters are then ligated to the fragments. Once ligation was assessed, the adapter-ligated product was PCR amplified. The final purified product was quantified using qPCR according to the qPCR Quantification Protocol Guide and qualified using the Caliper LabChipHigh Sensitivity DNA (PerkinElmer). For exome capture, 250 ng of DNA library was mixed with hybridization buffers, blocking mixes, RNase block and 5 μL of SureSelect all exon capture library, according to the standard Agilent SureSelect Target Enrichment protocol. Hybridization to the capture baits was conducted at 65 °C using heated thermal cycler lid option at 105 °C for 24 h on PCR machine. The captured DNA was then amplified. The final purified product was then quantified using qPCR according to the qPCR Quantification Protocol Guide, qualified using the TapeStation DNA screentape (Agilent) and then sequenced using the HiSeq 4000 platform (Illumina,San Diego, USA). Mapping and alignment were carried out as follows. First, read files (Fastq) were generated from the sequencing platform using the manufacturer's proprietary software. Mapping and alignment were carried out as follows. First, read files (Fastq) were generated from the sequencing platform using the manufacturer's proprietary software. Reads were mapped to their location in the reference human genome (GRCh38) using the Burrows-Wheeler Aligner (BWA) package, version 0.6.2. Duplicate reads were marked and removed using Samtools rmdup. Freebayes was used to call join variants on all samples and varaint frequencies were then extracted from the vcf file.

**Reporting summary**. Further information on research design is available in the Nature Research Reporting Summary linked to this article.

## Data availability

RNA-seq data has been deposited in NCBI's Gene Expression Omnibus (GEO) and is accessible using the accession number GSE134506. Whole exome sequencing data has been deposited in NCBI's Sequence Read Archive (SRA) and is accessible via SRA under the accession number PRJNA555070. All the other data supporting the findings of this study are available within the article and its supplementary information files and from the corresponding author upon reasonable request. A reporting summary for this article is available as a Supplementary Information file. The source data underlying Figs. 1–6 and Supplementary Figs. 1–11 are provided as a Source Data file

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

## Acknowledgements

We thank Leila Perie (Institut Curie, Paris), Shalin Naik (Walter and Eliza Hall Institute, Melbourne) Nuno Barbosa-Morais (iMM-JLA), Stefan Butz (DMMD, Zurich), Massimiliano Manzo (DMMD, Zurich) and Daniel Sobral (UCIBIO, Portugal) for technical advice; Inês Pereira, MD (Instituto Português de Oncologia–Francisco Gentil, Lisbon) for patient sample collection and the Animal, Bioimaging, Flow Cytometry facilities and the other members of the Silva-Santos lab (iMM-JLA) for support. This work was funded by the European Research Council (CoG_646701 to B.S.-S. and AdG Life-His-T to T.N.S) and Fundação para a Ciência e Tecnologia (SFRH/BPD/91344/2012 to F.C.; Welcome II programme, SFRH/BCC/105888/2014, SFRH/BPD/112968/2015 and EXPL/BIM/ONC/1656/2013 to H.N.; and UID/BIM/50005/2019 project funded by Fundação para a Ciência e a Tecnologia (FCT)/ Ministério da Ciência, Tecnologia e Ensino Superior (MCTES) through Fundos do Orçamento de Estado.

## Author contributions

F.C., H.N. and B.S-S. designed research and wrote the paper; F.C., H.N., D.M-S., C.J., N.S, T.C., C.R., R.F., B.K. and A.E.S. performed experiments and analyzed the data; T.B., C.R.V., M.G.S., M.G.M., and T.N.S provided key research tools; B.S-S. and H.N. supervised research.

## Competing interests

The authors declare no competing interests.
