## [Peer Review File · Nature Communications]

Reviewers' comments:

Reviewer #1 (Remarks to the Author):

Authors used two cell lines barcoded with lentiviral library, then treated with different chemotherapeutic agents and concluded, that whereas cells treated with Dox + Cyta (jargon as given by authors) led to accumulation of preexisting resistant clones, treatment with Dox + Cyta + DAC eliminated resistant clones. In addition authors performed RNAseq and exome sequencing on relapsing clones, also tried in vivo condition and patients AML samples, which, however, had only minor effect on entire story.

Not all details of the experiments are well described, (especially sequencing data and data processing) therefore it is not always easy to assess the credibility of given statements. In particular I do not see enough evidence of the claimed selection of preexisting resistant clone. My major concerns are as follows:

1. The size of the barcode library is not mentioned. Authors used 6×10^6 cell line cells for barcoding, which for 100% transduction efficiency requires at least 108 barcodes to make unique barcoding of each cell. The transduction efficiency is not known, the initial count of barcodes in the cell lines is not known. Therefore it is not clear what number of confirmed barcodes are detected in the cell lines before treatment and whether the barcodes really represent one cell derived clone or a mixture of those. Consequently, when we talk about clones, it is not clear whether they are real clones or mixture of clones marked with the same barcode.
2. It is expected (as it is known from other publications) that treating the barcoded cell population with chemo inhibitors should kill cells and reduce the number of clones (at least by chance). However, it was not systematically studied, data on barcode numbers and dynamics are very limited. There is sufficient number of recent publications in hematology field on barcoded cells regarding issues discussed above. None of those was cited in current paper.
3. The key claim that whereas treating with Doxo or Doxo+ Cyta gives persistently the same barcodes, but Doxo+Cyta+DAC generates "unpredictable" barcoded clones need statistical evidence. At present, authors only give the figure 3 as an evidence, which is not enough. There must be proper statistical analysis of the data. Authors should describe in details how many biological and technical replicates were done, what is the overall statistics and significance of that.
4. In addition to above, the experiment as it is performed now contains the caveat: authors compare 3 different treatments with three different survival levels. This difference is survival alone can generate the difference in the spectrum of recovered clones. Authors should repeat each treatment at the range of concentrations and select doses at which equal mortality/survival of clones is observed. Only at those conditions the data on survived clones can be compared.
5. More to the above. If treatment with Doxo and Doxo+Cyta selects preexisting resistant clones, then repeated treatment with the same drugs should give less mortality and more survival of the cells. In other words IC50 value for those drugs should increase after the first treatment. I might overlook it in the current version, but if it is not done, authors should do it to prove their claim.
6. After authors' claim that DAC have some special effect on the clonal (de)selection (although not really proven), they perform RNAseq and exome-seq, which would shed the light on this hypothetical effect. But it did not. The presented results are mainly limited to GSEA and GO plots, which helps to classify the genes, but is not a replacement of the entire analysis. What kind of adaptive strategy is revealed at the gene level? Authors could make more straightforward analysis on what gene changes they revealed compared to other publications on similar cell types at similar treatment. There are abundant publications on this subject. Why not do proper cross check own data and publications to compare the results and interpretations?

Minor notes:

7. Genome data has to be properly reported at least in the supplementary. In addition, the raw data should be submitted to GEO database or alike. Right now I could not find the reference number to any public database.
8. There is a repetitive reference to the stem cell signature, but not a clear explanation what it means. Markers CD34 and CD38 are very general for human stem cells and leukemic stem cells. The reference 44 refers to the epithelial cells, not hematopoietic one (at least by title). There are

more publications on genes specific to hematopoietic stem cells, which could be used for comparison. Authors could revisit their expression data and make more straightforward analysis on genes involved in self-renewal, in differentiation, in apoptosis, anything else related to the cell state and stemness. This would clarify the picture much better than GSEA plot.

Reviewer #2 (Remarks to the Author):

In this manuscript Caiado et. al. used lineage tracing to assess the clonal dynamics of human AML cell lines in the presence/absence of chemotherapeutic and demethylating agents. In a series of well-executed experiments, they showed that the addition of DNA methyltransferase inhibitors (DNMTi) prevents emergence of chemoresistant clones and drives expansion of chemosensitive clones. Following exome and transcriptome analyses of relapsed samples they show that addition of chemotherapy agents leads to relapses with quiescent phenotype whereas addition of DNMTi leads to more proliferative phenotype. Finally, they used transplantation experiments to conclude that chemotherapy selects pre-determined BC clones with stemness properties such as low proliferation, chemoresistance and leukaemia-initiating potential. In contrast, addition of DNMTi suppressed all the aforementioned properties of the clones.

This is a well-structured and written manuscript. However, I am not sure what is the novelty of authors discovery. As authors stated themselves the combination of low doses of DNMTi with standard chemotherapy regimens have shown clinical benefit. In addition, it has been shown that low proliferation and quiescent states in leukemic blasts are associated with resistance to cytotoxic drugs, as chemotherapeutic agents preferentially target dividing cells. Finally, it has been demonstrated that during myeloid differentiation, DNA methylation changes occur predominantly in the direction of hypomethylation (doi: 10.1182/blood-2013-02-482893, doi: 10.1038/nature09367, etc.) so it is not surprising that addition of hypomethylating agent leads to reduced frequency of leukaemia stem cells i.e. pushes cells toward differentiation. The authors should make it clearer as to what is the contribution of their study/findings in the light of the existing knowledge.

Reviewer #3 (Remarks to the Author):

Caiado et al. employ a barcoding strategy in two leukemia cell lines, HEL and OCI-AML3, to track clonal competition and selection of chemoresistant clones in response to clinically relevant anthracycline+nucleoside analog combo with and without hypomethylating agent (decitabine, DAC). They find that doxorubicin strongly selects predetermined chemoresistant clones, while addition of DAC reduces the number of selected clones and chemoresistance. Targeted DNA sequencing does not detect additional mutations that can explain chemoresistance (potentially suggesting an epigenetic mechanism). The authors go on to perform transcriptome studies that they interpret as gain of cancer "stemness" after treatment with Dox+Ara-C, which is abrogated by DAC. The results with the stem cell immunophenotype are mostly replicated in primary AML samples.

Overall this is an elegant approach to an important clinical issue, namely, development of chemoresistance and resultant AML relapse. The use of barcoding to follow individual clones in a complex population is potentially very powerful. The study will be of interest to a wide audience of cancer biologists. Overall the paper is well-written, although it would benefit from a more detailed explanation of some of the approaches. As one example, explanation how individual BC fitness was determined (relevant to Figure 3) would be helpful. Another suggestion would be to include DEseq tables for gene expression studies, and lists of genes used for GSEA analysis other than standard MSigDB signatures (relevant to Figure 5).

My biggest concern is the lack of epigenetic profiling studies in the paper. Since the mechanism of chemoresistance, and its reversal by DAC, is likely epigenetic, I feel this is a missed opportunity. Although single-cell approaches would be ideal for heterogeneous cell mixtures, they may be too costly and impractical. Therefore, even some bulk broad-stroke analyses such as H3K27ac vs H3K27me3 ChIP-seq, preferably combined with DNA methylation profiling, would be informative. In addition, the mechanistic part currently in the manuscript describing “stemness” and cell proliferation (Figures 5 and 6) only scratches the surface and needs to be fleshed out.

Specific comments:

1. Individual BC fitness (Fig 3): here it is defined as fold increase over untreated control. If a certain drug combo kills a larger proportion of cells, and clones, the fold enrichment of surviving BCs is higher. However, this does not necessarily mean these enriched clones have higher fitness, ie ability to survive 3 drugs rather than 2. To use an analogy, those athletes who can finish a marathon do not necessarily run it faster than 10K just because more people were able to finish a 10K than a marathon. Please clarify.
2. Alternative interpretation of data: The authors state that “DAC combination suppresses the re-growth of chemoresistant clones, while favoring the selection and massive expansion of rarer and less fit clones that remain sensitive to chemotherapy re-treatment”. As alternative interpretation of the same data may be that Dox favors clones with a pre-existing gene expression profile, while DAC being an epigenetic reprogrammer leads to acquisition of a new epigenetic profile in random clones upon treatment. Have you looked at epigenetic profiles of outgrowing clones to test if this is the case? This would be consistent with a low correlation between BC composition = stochasticity in DAC-treated replicates. At the same time, Fig. 5h shows outgrowth of roughly the same clones after DAC as those that were chemoresistant in Dox+AraC. Can you please elaborate on the relationship between this finding and previous results?
3. The basic pharmacology/pharmacogenomics of Dox and Ara-C sensitivity is reasonably well studied. Sensitivity to Dox is driven by ABC transporters and TopoII, among other things. Ara-C is determined by drug metabolism, there are SNPs associated with favorable/unfavorable clinical responses. Have you checked this, including in your gene expression data?
4. Acquisition/loss of quiescence: This part needs more experimental evidence to fully make this claim. One gene (CDKN1C) is not enough. Doubling time is a measure of a combination of factors, including the ability of cells to survive dilution/lower density at each passage and removal of conditioned media that might contain favorable cytokines produced by neighboring cells. Detailed cell cycle analysis would be necessary, including % Ki67+, BrdU incorporation, and cell cycle duration by cell synchronization, or number of cell divisions by label dilution.
5. There seems to be a problem with Figure S7 – some panels are not described in the text, others seem to be out of context, while panels e-f do not exist.
6. Need more details on efflux activity measurements by flow cytometry of a fluorescent substrate. First, doxorubicin is itself fluorescent; it is unclear if this was controlled for. The use of ABC inhibitors would be a good additional control of specificity. Second, there are multiple transporters with expression levels readily assessable by mRNA abundance, please check your RNA-seq data.
7. “Data not shown” – “Doxo+Cyta+DAC chemosensitive cells showed reduced L-IC frequency...” Please show the data, even if it has to go to the supplement.
8. Stem cell gene expression signature in Fig 5a: How was this signature selected? What genes were included? In its present form, there are too many genes in the reference signature (too many hits in the GSEA plot), which is making this less meaningful while artificially driving the p-value down. There are published HSC and LIC signatures in the literature, including the 17-gene signature cited in this paper. It would be informative to compare your gene set against some of those.
9. Figure S8: In panel b, there is no difference between Dox/AraC and Dox/AraC/DAC. Panel c: can you please clarify what exactly was measured?
10. Figure 6: In vivo studies were only conducted using HEL cell line. What about OCI-AML3? This is important because OCI-AML3 carry a DNMT3A mutation, which may impact DAC sensitivity. In addition, a DAC-only arm would be a good control here. Panel g (primary AML samples): when

authors state there is a decrease in LSC frequency in 6 out of 8 tested samples, which samples are those exactly? Please clarify.

11. Discussion: "we found that ... there are hierarchically related cells (BC-clone) that already share the molecular ..." There is no data in the manuscript to support the statement regarding clonal hierarchies. Please rephrase.

12. Statistics: one-way ANOVA is used for pairwise comparisons throughout the paper. Why was this method chosen? A t-test (for normally distributed data) and non-parametric tests (for non-normally distributed data) are common for these types of comparisons. Please clarify.

Reviewer #4 (Remarks to the Author):

Caiado and colleagues present an interesting platform for evaluating patterns of clonal evolution as a mechanism of drug resistance that involves introducing single-cell-specific, molecular barcodes into AML-derived cell lines, exposure of cells to drugs or drug combinations and then comparison of barcode frequencies at time 0 and after cells have recovered. This enables them to study diversity of clonal selection after drug exposure as well as the stability of outgrowth of specific founder cells in response to drug regimens. They find that the inclusion of hypomethylating agents plus doxorubicin and cytarabine leads to less diversity of clones and cells that are still sensitive upon re-challenge (in contrast to doxo + cyta without hypomethylating agents). This corresponds with changes in patterns of gene expression that maintain cells in a proliferative state rather than the doxo+cyta alone transcriptional program response, which slows cell proliferation rate and converts cells into a more primitive differentiation state. The findings are interesting and potentially compelling for new therapeutic targets that may prevent outgrowth of AML cells that are resistant to a commonly used chemotherapy regiment. There are a number of additional lines of work that would strengthen this study:

1. The statement that chemo + DAC eliminates chemoresistance cells is overstated. To demonstrate this, the authors need to re-challenge the cells and show that nothing can grow back out.

2. It would also be important to show full dose response curves of the non-treated cells versus each treatment condition for this re-challenge experiment.

3. What happens to BC frequencies after a second "salvage" round of "treatment"?

4. The xenograft model of HEL cells is not convincing and the reported toxicity is concerning for eventual clinical application of this strategy

5. The use of surface markers to define LSCs is not phenotypically-defined LSCs as claimed. How do you know these are LSCs and not healthy mono/myeloblasts?

6. The improvement of doxo/cyta/DAC over doxo/cyta in patient samples only really looks convincing for 2/8 cases.

7. Based on the findings and the narrative, it would be important to target CDKN1C either genetically or pharmacologically and show that this reduces resistance to doxo + cyta.

8. There are several large, publicly available AML datasets (TCGA, Beat AML, etc.). It would be useful to look at CDKN1C expression versus mutational/prognostic groups in these datasets.

9. It is often not clear which cell line is being used for data in the paper figures, particularly in the main paper figures after figure 1. It seems as though much of the data was generated from a single cell line (HEL), though some of it may have been replicated in OCI-AML3 in the supplement.

It will be important to be very clear which line is being used for every experiment and finding. For instance, for the finding of down-regulated CDKN1C with doxo/cyta/DAC, has this been validated in both cell lines or only in HEL?

Point-by-point reply to the Reviewers of NCOMMS-19-04936 (Caiado et al.)

We thank the Reviewers for the constructive criticism and suggestions to improve our manuscript by providing additional experimental data to support our claims. We have performed a series of experiments whose results have been added in **revised versions of Figures 1-6** and **Supplementary Figures 8 and 9**, and the **new Supplementary Figures 1, 4 and 10**, and which are discussed below in our point-by-point reply.

Reviewer #1:

Major Comments:

1. *The size of the barcode library is not mentioned. Authors used 6x10⁶ cell line cells for barcoding, which for 100% transduction efficiency requires at least 108 barcodes to make unique barcoding of each cell. The transduction efficiency is not known, the initial count of barcodes in the cell lines is not known. Therefore it is not clear what number of confirmed barcodes are detected in the cell lines before treatment and whether the barcodes really represent one cell derived clone or a mixture of those. Consequently, when we talk about clones, it is not clear whether they are real clones or mixture of clones marked with the same barcode.*

We have added information relative to the barcode library size, transduction efficiency and initial count of barcodes in the cell lines in the **Materials and Methods section (page 18 – line 20 and page 19-20, lines 25-33, 1-5)**. Briefly, using a library with ~2600 barcodes, we transduced cells at <5% efficiency (as to limit presence of cells transduced with more than 1 barcode) and limited use to sublines carrying ~10% of the library's total barcode repertoire (to minimize risk of an individual barcode representing >1 initial cell/lineage). Under these conditions, and based on technical controls performed in line with those described in previous publications by co-author Ton Schumacher that are cited in our study (*Schepers et al. J Exp Med 2008; Naik et al. Nature 2013*), we ensure that each cell progeny we trace carries a single barcode that is uniquely represented by this lineage. These restrictive conditions allow us to assume with high confidence that each barcode marks a real clone (i.e., a group of cells derived from a single cell, transduced with a unique barcode).

2. *It is expected (as it is known from other publications) that treating the barcoded cell population with chemo inhibitors should kill cells and reduce the number of clones (at least by chance). However, it was not systematically studied, data on barcode numbers and dynamics are very limited. There is sufficient number of recent publications in hematology field on barcoded cells regarding issues discussed above. None of those was cited in current paper.*

We thank the Reviewer for pointing this out. As suggested, we have included and briefly discussed in the **Introduction (page 4 – lines 16-21)** a key study (*Bhang et al. Nat Med 2015*) that used a barcoding strategy to trace the origin of resistance to targeted therapies to rare pre-existing clonal populations carrying specific mutations in the genes targeted by the specific therapy used – in this case, EGFR and ABL1 inhibitors in *in vitro* models of non-small cell lung cancer and chronic myeloid leukemia, respectively.

3. *The key claim that whereas treating with Doxo or Doxo+ Cyta gives persistently the same barcodes, but Doxo+Cyta+DAC generates “unpredictable” barcoded clones need statistical evidence. At present, authors only give the figure 3 as an evidence, which is not enough. There must be proper statistical analysis of the data. Authors should describe in details how many biological and technical replicates were done, what is the overall statistics and significance of that.*

Our claim that Doxo or Doxo+Cyta treatment selects for the same barcodes and that Doxo+Cyta+DAC selects for unpredictable barcodes derives from multiple analyses now displayed in the **new versions of Figure 2**. In Figure 2d (equal to previous) we have performed multiple correlations between the barcode compositions of each technical replicates of each 1 of 3 independent experiments, for a total of 8-10

technical replicates (now clarified in figure legend), and observed that Doxo or Doxo+Cyta have average Pearson correlations of 0.9 and 0.4 respectively, indicating that replicates within each group are strongly or moderately similar to each other. In contrast, Doxo+Cyta+DAC replicates have a much lower average Pearson correlation of approximately 0.1, which indicates that on average the replicates in this group are completely different from each other. Next, in the **new Figures 2e** and **2g**, we determined which barcodes are resistant to each therapy. This was determined by calculating the fold variations of each BC-clone between T0 and Trelapse in each treatment group compared to NT (technical replicates in each independent experiment are now discriminated in the figure). Therapy-resistant BC-clones were defined as the ones not showing a statistical significant decrease in fold variation relative to NT (schematically represented in the left panel of Figure 2e; the statistical calculations consisted of multiple t-tests, one per barcode, $p < 0.05$). By overlapping therapy resistant BCs across all independent experiments (now discriminated in Figure 2e), we identified 5 BCs and that are consistently selected by Doxo and Doxo+Cyta, and thus pre-determined to relapse to these treatments (clarified in **page 7, lines 19-22**). We have now included the same analysis performed on Doxo+Cyta+DAC relapses in the **new Figure 2g** (discussed in **page 7, lines 28-30**) where we unequivocally show that there are no common BCs resisting Doxo+Cyta+DAC in 3 independent experiments, thus confirming the unpredictability of the barcodes relapsing to this treatment.

- 4. The experiment as it is performed now contains the caveat: authors compare 3 different treatments with three different survival levels. This difference in survival alone can generate the difference in the spectrum of recovered clones. Authors should repeat each treatment at the range of concentrations and select doses at which equal mortality/survival of clones is observed. Only at those conditions the data on survived clones can be compared.***

This is a very important point. However, it is crucial to note that the survival levels in fact were equal between Doxo+Cyta and Doxo+Cyta+DAC (Fig 1b), only Doxo shows higher survival compared to the other 2 treatments. To address this we have now done, as suggested by the Reviewer, drug titrations within each treatment and compared the effects on BC-clonal composition at Trelapse in drug doses that caused equivalent cellular mortality. As shown in **new Supplementary Figure 4** and discussed on **page 7 (lines 3-17)**, we establish that in all treatments, maximum measured cell death levels correlate positively with BC-clone elimination. Importantly we nevertheless observe that this effect is strongest (highest Pearson correlation, lower p-value) in Doxo+Cyta+DAC treatment, suggesting that this treatment needs to kill fewer cells to achieve the same degree of BC-clone elimination as the other treatments. Furthermore, by comparing 1x dose of Doxo+Cyta and Doxo+Cyta+DAC to a 9x higher dose of Doxo, we observe equivalent cell elimination levels but the degree of BC-clone elimination is clearly higher in Doxo+Cyta+DAC, and only in this group we observe a lack of chemoresistance development, which is fully consistent with our initial observations.

- 5. If treatment with Doxo and Doxo+Cyta selects preexisting resistant clones, then repeated treatment with the same drugs should give less mortality and more survival of the cells. In other words IC50 value for those drugs should increase after the first treatment. I might overlook it in the current version, but if it is not done, authors should do it to prove their claim.***

We have performed IC50 value determination for doxorubicin in Trelapse samples from HEL and OCIAML3 cell lines. As shown in **new Figure 1d** and discussed in **page 6 (lines 1-3)**, we observe a significant 2-4 fold increase in IC50 values of Doxo and Doxo+Cyta Trelapse samples compared to NT, confirming that these relapsing samples develop resistance to doxorubicin. On the other hand, Doxo+Cyta+DAC show no increase in the IC50 values thus confirming the lack of chemoresistance acquisition in these Trelapse cells, as previously shown (Figure 1c). We thank the Reviewer for his/ her suggestion that led to this additional experimental support of our initial claim.

- 6. After authors' claim that DAC have some special effect on the clonal (de)selection (although not really proven), they perform RNAseq and exome-seq, which would shed the light on this hypothetical effect. But it did not. The presented results are mainly limited to GSEA and GO plots, which helps to classify the genes, but is not a replacement of the entire analysis. What kind of adaptive strategy is revealed at the gene level? Authors could make more straightforward analysis on what gene changes they revealed compared to other publications on similar cell types at similar treatment. There are abundant publications on this subject. Why not do proper cross check own data and publications to compare the results and interpretations?***

We respectfully disagree with the Reviewer on this point, since we consider that the RNA-seq data in the original version of the manuscript clearly reflected the effects of DAC on clonal selection. In detail, the

increase in proliferative/ biogenesis pathways in this group strongly supported our observation that Doxo+Cyta+DAC relapses showed increased clonal expansions (of rarer and unfit BC-clones), which outcompeted chemoresistant BC-clones. This data interpretation was further highlighted in the new version of the manuscript (**page 10, lines 31-32**) and was kept in the discussion (page 17). This notwithstanding, to address the adaptive strategy at the gene level as suggested by the Reviewer, we have now included in a **new Figure 4d panel** (described on **page 11, lines 2-11**) in which we discriminate the genes associated to quiescence/proliferation that show significant differential expression between Doxo+/-Cyta and Doxo+Cyta+DAC. Moreover, to further address the chemoresistance phenotype, we performed a straightforward analysis of genes that have been previously implicated in doxorubicin/cytarabine resistance (including a new reference on this subject). We observed that some gene members of the ABC transporter family were overexpressed in chemoresistant (Doxo+/-Cyta) compared to chemosensitive (Doxo+Cyta+DAC) relapses, which resulted in increased efflux capacity of chemoresistant relapses. This data was included in the **new Figure 4h and i panels** and described on **page 11, lines 21-33**.

Minor Comments:

7. *Genome data has to be properly reported at least in the supplementary. In addition, the raw data should be submitted to GEO database or alike. Right now I could not find the reference number to any public database.*

Exome data has now been properly reported in **Supplementary Figure 8a**, where we include general parameters related to the read quality and mapping such as: number of mapped reads (% total reads), number of mapped bases (% total bases), average read length, target regions coverage, average read depth (target regions). Both RNA-seq and Exome seq data have now been submitted to GEO data base, and the respective reference/ access numbers are indicated in the **Material and Methods** section.

8. *There is a repetitive reference to the stem cell signature, but not a clear explanation what it means. Markers CD34 and CD38 are very general for human stem cells and leukemic stem cells. The reference 44 refers to the epithelial cells, not hematopoietic one (at least by title). There are more publications on genes specific to hematopoietic stem cells, which could be used for comparison. Authors could revisit their expression data and make more straightforward analysis on genes involved in self-renewal, in differentiation, in apoptosis, anything else related to the cell state and stemness. This would clarify the picture much better than GSEA plot.*

The adult-tissue stem cell signature was selected based on a previous publication by *Milanovic et al. Nature 2018*, where it was shown that chemotherapy-induced senescence reprograms cells towards increase stemness properties (leading to enrichment of this specific signature). This reference and a brief clarification of its meaning was included on **page 12, lines 16-17**. Considering the Reviewer's concern with the high number of genes in this signature, we have now included on **new Figure 5a panel** (described in **page 12, lines 15-18**) additional stem cell signatures (hematopoietic, mesenchymal, cancer and leukemic) that are enriched in chemoresistant compared to chemosensitive relapses. Also following the Reviewer's advice, we have performed a straightforward comparison of genes implicated in HSC and LSC self-renewal (key stem-cell property) that show significant differential expression between Doxo+/-Cyta and Doxo+Cyta+DAC, which is now included in **new Figure 5b panel** and described on **page 12, lines 18-21**.

Reviewer #2:

Major Comments:

The authors should make it clearer as to what is the contribution of their study/findings in the light of the existing knowledge.

The novelty of our study relates to the description of how the clonal dynamics that exist beyond the level of genetics underpins chemoresistance development in hAML relapse to standard chemotherapy, but perhaps most

importantly how this dynamic is impacted by the combination with hypomethylating drugs - leading to clarification of the underlying mechanisms behind a decreased chemoresistance development. To stress this point we have altered the last sentence of the **Introduction (pages 5-6)**.

Reviewer #3:

Major Comments:

1. *Individual BC fitness (Fig 3): here it is defined as fold increase over untreated control. If a certain drug combo kills a larger proportion of cells, and clones, the fold enrichment of surviving BCs is higher. However, this does not necessarily mean these enriched clones have higher fitness, ie ability to survive 3 drugs rather than 2. To use an analogy, those athletes who can finish a marathon do not necessarily run it faster than 10K just because more people were able to finish a 10K than a marathon. Please clarify.*

Individual BC fitness is clearly defined as “the fold change of the frequency of each individual BC-clone between T0 and Trelapse in NT conditions” (**page 8 lines 12-13**) and **not over NT control** as stated by the Reviewer. According to our definition, fitness defines the intrinsic capacity of any given BC-clone to expand (up to the time of Trelapse, defined in parallel using the treated groups) in the absence of any therapy (NT conditions). Our claim is that the BC-clones that relapse to Doxo+Cyta+DAC are the ones with lower fitness, which means that, under neutral (NT) conditions, these BC-clones are outcompeted over time by fitter BC-clones. Clarification of the definition was highlighted on **Fig3e panel**.

2. *Alternative interpretation of data: The authors state that “DAC combination suppresses the re-growth of chemoresistant clones, while favoring the selection and massive expansion of rarer and less fit clones that remain sensitive to chemotherapy re-treatment”. As alternative interpretation of the same data may be that Dox favors clones with a pre-existing gene expression profile, while DAC being an epigenetic reprogrammer leads to acquisition of a new epigenetic profile in random clones upon treatment. Have you looked at epigenetic profiles of outgrowing clones to test if this is the case? This would be consistent with a low correlation between BC composition = stochasticity in DAC-treated replicates. At the same time, Fig. 5h shows outgrowth of roughly the same clones after DAC as those that were chemoresistant in Dox+AraC. Can you please elaborate on the relationship between this finding and previous results?*

The Reviewer raises the very interesting hypothesis that DAC modifies the epigenetic profiles of relapsing clones, changing their gene expression in a somewhat random fashion leading to the observed outgrowth of unpredictable BC-clones. To address this hypothesis we have preformed CHIP-qPCR in bulk Trelapse samples for the active/inactive histone3 marks K4me3 and K27me3 in genes that are differentially expressed between Doxo+Cyta and Doxo+Cyta+DAC groups: *ABCA2; ABCC3; CDKN1C; CDKN1B; TXNIP; FOXO1; HOXA4; ANG* (all higher in Doxo+Cyta) and *E2F1; MYC; MET* (all higher in Doxo+Cyta+DAC). As shown in the **Reply Figure to Reviewer 3, issue 2 panel a** displayed below, we observe that the majority of genes upregulated in Doxo+Cyta (*ABCA2; ABCC3; CDKN1C; CDKN1B; TXNIP; FOXO1*) show increased active K4me3 compared to NT, which is strikingly reduced upon DAC combination. This strongly suggests that DAC (likely via an indirect mechanism) interferes with the Doxo+Cyta induced K4me3 marking, thus affecting the active transcription of these genes. However, we do not observe a similar pattern on the genes upregulated in the Doxo+Cyta+DAC group (*E2F1; MYC; MET*). Additional analysis of promoter CpG island methylation of *E2F1* and *MYC* genes in Trelapse samples (**panel b**) shows an overall unmethylated state in all conditions, thus precluding a direct methylation-dependent regulation of these genes at Trelapse. These observations suggest that although gene expression regulation via histone3 methylation might be an epigenetic mechanism underlying our biological observations, its detailed clarification poses a high degree of complexity that detracts from exploring it further in the current manuscript. As noted by the Reviewer, the proper assessment of an epigenetic mechanism in our setting should be preformed using single-cell approaches in a longitudinal fashion which is “costly and impractical” – thus better fitting in follow-up studies.

Also, we observe a very interesting phenomenon in the *CDKN1C* gene, which is the existence of a bivalent K4me3 and K27me3 state in chemoresistant cells. This may suggest that cell populations that are exposed to chemotherapy keep *CDKN1C* gene under a flexible transcriptional control that allows, on one hand, that cells continue to proliferate in the absence of chemotherapy (lowering *CDKN1C* expression); but, on the other hand, still keep the “epigenetic memory” of previous exposure to chemotherapy to induce fast *CDKN1C* expression upon re-exposure to chemotherapy, leading to cell-

cycle blockade and prompt entry into quiescence. Taking these speculations into consideration, we believe that this data set would be more relevant if properly developed and channeled into a follow-up story concerning the epigenetic regulation of CDKN1C upon chemotherapy exposure. Concerning the Reviewer's comment on Figure 5h, the key aspect to consider is that we are scoring presence (+, green) or absence (-, red) of BC-clones and not just a relative decrease in the frequency as shown previously (Figure 3c). We still consider that there is a clear difference on the BC-clones present in Doxo+Cyta and Doxo+Cyta+DAC, particularly in the first half of the table (all are absent in Doxo+Cyta+DAC, as indicated by red color compared to a more mixed pattern in Doxo+Cyta).

Reply Figure to Reviewer 3, issue 2. ChIP-qPCR analysis of Trelapse samples for active/ inactive histone H3 marks K4me3/ K27me3, in genes differentially expressed between Doxo+Cyta and Doxo+Cyta+DAC groups.

3. *The basic pharmacology/pharmacogenomics of Dox and Ara-C sensitivity is reasonably well studied. Sensitivity to Dox is driven by ABC transporters and TopoII, among other things. Ara-C is determined by drug metabolism, there are SNPs associated with favorable/unfavorable clinical responses. Have you checked this, including in your gene expression data?*

To address this issue, we performed a straightforward analysis of genes that have been implicated in doxorubicin / cytarabine resistance. We observed that some gene members of the ABC transporter family were overexpressed in chemoresistant (Doxo+/-Cyta) compared to chemosensitive (Doxo+Cyta+DAC) relapses, which resulted in increased efflux capacity of chemoresistant relapses. These data have been included in **new Figure 4h and i panels** and described on **page 11, lines 21-33**. Further following reviewers suggestion we have now incorporated Supplementary Data 1 file displaying differential gene expression lists between the different experimental groups.

4. *Acquisition/loss of quiescence: This part needs more experimental evidence to fully make this claim. One gene (CDKN1C) is not enough. Doubling time is a measure of a combination of factors, including the ability of cells to survive dilution/lower density at each passage and removal of conditioned media that might contain favorable cytokines produced by neighboring cells. Detailed cell cycle analysis would be necessary, including % Ki67+, BrdU incorporation, and cell cycle duration by cell synchronization, or number of cell divisions by label dilution.*

This point has been addressed by multiple approaches. As stated by the Reviewer, one gene is not enough to claim the changes in proliferation. To address this we have now extended our analysis to multiple genes implicated in quiescence/proliferation that show significant differential expression between Doxo+/-Cyta and Doxo+Cyta+DAC, which are now included in **new Figure 4d panel** (described on **page 11, lines 2-11**). We have further performed cell-cycle analysis (Ki67, DAPI) and cell division history by CFSE label dilution. The data displayed in the **new Supplementary Figures 10a,b,c** and described on **page 11, lines 11-15** clearly show that cells relapsing to Doxo+Cyta+DAC have, additionally to decreased doubling times, increased G2-S-M cell cycle stage frequency (**Fig S10a,b**) and lower fractions of undivided cells at all time points (**Fig S10c**) when compared to Doxo±Cyta relapses.

5. *There seems to be a problem with Figure S7 – some panels are not described in the text, others seem to be out of context, while panels e-f do not exist.*

We thank the reviewer for pointing this out. Figure S7 is now **Supplementary Figure 9**. The mistakes in the previous figure calling in the text have been corrected in the revised version of the manuscript.

6. *Need more details on efflux activity measurements by flow cytometry of a fluorescent substrate. First, doxorubicin is itself fluorescent; it is unclear if this was controlled for. The use of ABC inhibitors would be a good additional control of specificity. Second, there are multiple transporters with expression levels readily assessable by mRNA abundance, please check your RNA-seq data.*

ABC transporter activity was analyzed using the eFlux-ID Gold multidrug resistance kit (Enzo Life Sciences) according to the manufacturer's instructions, as stated in the **Material and Methods** section. The cells used in this assay were Trelapse samples collected 25-30 days after direct doxorubicin exposure and continuously cultured with periodical changes of media, therefore reducing very significantly the possibility of doxorubicin fluorescence contamination. Following the Reviewer suggestion, we checked for ABC gene family expression in Trelapse samples and added **new Figure 4h and i panels** (described on **page 11, lines 21-33**.) where we show that various members of the ABC transporter family that have been implicated in AML relapse to chemotherapy (*ABCA2, B6 and C3*) were significantly overexpressed in chemoresistant (Doxo+/-Cyta) compared to chemosensitive (Doxo+Cyta+DAC) relapses, which associated in increased efflux capacity of chemoresistant relapses. Given the involvement of multiple ABC genes (some of which have no available commercial inhibitors) we have not included testing of the effect of ABC inhibitors in our manuscript.

7. *“Data not shown” – “Doxo+Cyta+DAC chemosensitive cells showed reduced L-IC frequency...” Please show the data, even if it has to go to the supplement.*

The survival data on mice transplanted with samples from NT, chemoresistant (Doxo+/-Cyta) and chemosensitive (Doxo+Cyta+DAC) relapses has now been included as new **Figure 5g** (described on **page 12, lines 31-32**).

8. *Stem cell gene expression signature in Fig 5a: How was this signature selected? What genes were included? In its present form, there are too many genes in the reference signature (too many hits in the GSEA plot), which is making this less meaningful while artificially driving the p-value down. There are published HSC and LIC signatures in the literature, including the 17-gene signature cited in this paper. It would be informative to compare your gene set against some of those.* SAME POINT RAISED BY REVIEWER 1 - COMMENT 8.

The adult-tissue stem cell signature was selected based on previous publication by *Milanovic et al. Nature 2018* where it was shown that chemotherapy-induced senescence reprograms cells towards increase stemness properties (leading to enrichment of this specific signature). This reference and a brief clarification of its significance were included on page 12, lines 16-17. Considering the Reviewer's concern with the high number of genes in this signature, we have now included on **new Figure 5a panel** (described in page 12, lines 15-18) additional stem cell signatures (hematopoietic, mesenchymal, cancer and leukemic) that are enriched in chemoresistant compared to chemosensitive relapses. Additionally, we have performed a straightforward comparison of genes implicated in HSC and LSC self-renewal (key stem-cell property) that show significant differential expression between Doxo+/-Cyta and Doxo+Cyta+DAC, which is now included in **new Figure 5b panel** (described in page 12, lines 18-21).

9. *Figure S8: In panel b, there is no difference between Dox/AraC and Dox/AraC/DAC. Panel c: can you please clarify what exactly was measured?*

This is now **Supplementary Figure 11** in the revised version. In panel b we quantified the number of live cells over time upon *ex vivo* exposure to Doxo+Cyta and Doxo+Cyta+DAC. We indeed observed an identical level of cell survival, which is in agreement with our previous *in vitro* observations (Figure 1b). This panel b simply sets the stage to look at Trelapse BC-clones derived from a leukemia initiating cell (established *in vivo*, see panel a), termed L-IC BC-clones. In panel c, which contains the key finding in this figure, we quantified the number of BC detected in these *ex vivo* treated Trelapse samples and normalized it to the number of L-IC BC-clones observed in untreated (NT) *ex vivo* cultured cells. This confirmed the enhanced ability of DAC combination to target L-ICs, which we hope is clear from the data description on page 13, lines 12-17.

10. *Figure 6: In vivo studies were only conducted using HEL cell line. What about OCI-AML3? This is important because OCI-AML3 carry a DNMT3A mutation, which may impact DAC sensitivity. In addition, a DAC-only arm would be a good control here. Panel g (primary AML samples): when authors state there is a decrease in LSC frequency in 6 out of 8 tested samples, which samples are those exactly? Please clarify.*

The Reviewer makes an interesting point regarding the potential impact of the DNMT3a mutation R882 carried by the OCIAML3 cell line, on the effects of DAC combination treatment. Of note, we do observe that OCIAML3 cells are more sensitive to cytarabine and less sensitive to doxorubicin, which is in agreement with a recent study by *Guryanova et al. Nat Med 2016*, showing that *DNMTA R882* mutations confer resistance to anthracycline drugs. However, we do not find an additional *in vivo* model justified since the comprehensive *in vitro* data set we provide on OCIAML3 suffice to establish that this DNMT3a mutation does not have an obvious impact on the major phenotypes/ clonal dynamics observed.

Another reason why we think that performing *in vivo* treatment of OCIAML3 would not be helpful for our study is that, as discussed in the manuscript, the doses of chemotherapy used *in vivo* are restricted by the toxicity of the mouse model, and therefore need to be significantly reduced, which severely limits the data interpretation.

As suggested by the Reviewer, we have added a DAC-only arm as control in the **new Figure 6b and c** panels (described in page 14, line 3) and observed that, as expected, DAC-alone treated mice show a mild reduction in cell number in peripheral blood and a small increase in overall survival.

With regard to the Reviewer's question on which primary human samples show reduced LSC levels we have a **new Figure 6g** panel in which we quantify the absolute number of LSCs 6days after treatment in NT, Doxo+Cyta and Doxo+Cyta+DAC (described in page14, lines 21-25). We observed a significant reduction on LSC absolute numbers in the following 6 out of 8 samples (labels): 1, 3, 12, 15, 20 and 25.

11. *Discussion: "we found that ... there are hierarchically related cells (BC-clone) that already share the molecular ..."* There is no data in the manuscript to support the statement regarding clonal hierarchies. Please rephrase.

The statement was rephrased to “there are clonally related cells” (**page16, line30**)

- 12. Statistics: one-way ANOVA is used for pairwise comparisons throughout the paper. Why was this method chosen? A t-test (for normally distributed data) and non-parametric tests (for non-normally distributed data) are common for these types of comparisons. Please clarify.**

One-way ANOVA was used for multiple group comparisons and not for pairwise comparisons as stated. t-test and Mann-Whitney testing were used for pairwise comparisons when necessary and as described.

Reviewer #4:

Major Comments:

- 1. The statement that chemo + DAC eliminates chemoresistance cells is overstated. To demonstrate this, the authors need to re-challenge the cells and show that nothing can grow back out.**
- 2. It would also be important to show full dose response curves of the non-treated cells versus each treatment condition for this re-challenge experiment.**
- 3. What happens to BC frequencies after a second “salvage” round of “treatment”?**

These are all important points. To address point #1 we have, as suggested by the Reviewer, re-exposed Trelapse samples from the different treatment groups to a new round of chemotherapy (at the same concentration as in the first round) – Doxo cells were re-exposed to Doxo; Doxo+Cyta and Doxo+Cyta+DAC cells were re-exposed to Doxo+Cyta. Cells numbers were quantified over time and the number of replicates in each group that achieved a second relapse (R2) was quantified. We also included non-treated cells to address point #2. While NT samples behaved as previously shown (Fig1b), with 100% of treated replicates relapsing, as shown in **new Supplementary Figure 1** (panels **a to c** - described on **page 6, lines 3-12**), whereas 100% (3 out of 3 Doxo, 4 out 4 Doxo+Cyta) Doxo±Cyta Trelapse samples relapsed a second time, only 25% (1 out of 4) of the Doxo+Cyta+DAC Trelapse samples achieved relapse within the same time frame upon re-treatment. This data thus confirms that DAC combination largely prevents chemoresistance development in our system.

Concerning point #3, as shown in the **new Supplementary Figure 1** (panels **d to g** - described on **pages 7-8**) we have assessed the BC numbers and architectures at first and second relapse (R1 and R2) and confirmed that although R2 samples showed reduced BC-clone number compared to R1 samples, their BC architecture was highly correlated with R1, suggesting a conserved BC composition. In fact, by comparing the summed frequency of the Doxo±Cyta resistant BC-clones in R1 and R2, we observed an increase at R2 with clonal expansions of individual Doxo±Cyta resistant BC-clones. Concerning the only Doxo+Cyta+DAC replicate that achieved a second relapse (**Supplementary Figure 1 panel g, lower sample**), we observed that this sample represented an exception where chemoresistant BC-clones were still present at R1 and strikingly were further expanded at R2, further confirming their chemoresistant properties.

- 4. The xenograft model of HEL cells is not convincing and the reported toxicity is concerning for eventual clinical application of this strategy.**

There are currently no convincing in vivo models for chemotherapeutic treatment of hAML xenografts, i.e. protocols that allow for strong elimination of established disease/ induce remission (or of subsequent recurrence) as observed in AML patients. This results from the fact that in vivo establishment of human AML cells in mouse models has to be preformed in sublethally irradiated immunocompromised mice that show increased sensitivity to chemotherapeutic drugs (particularly to anthracycline drugs like doxorubicin), thus limiting the dosage of drugs and consequently the capacity of inducing remissions in these models. Notably, we did not observe additional toxicity from adding DAC to the chemotherapy regimen in our mice, so we have no reason to expect this combination to be particularly complicated from a toxicity point of view. Most importantly, several clinical studies combining low doses of DAC with standard chemotherapy regimens have in fact already been performed (REF 25-27), which strongly supports the clinical translatability of our findings.

Taking these observations into consideration (discussed in our manuscript, **page 14, lines 1-2 and discussion**), the in vivo data generated for this study is a proof-of-concept of that combination with DAC prevents development of chemoresistance in human AML cells exposed to chemotherapy, rather than aimed to act as a direct blueprint for clinical application of this regimen.

5. *The use of surface markers to define LSCs is not phenotypically-defined LSCs as claimed. How do you know these are LSCs and not healthy mono/ myeloblasts?*

To address the Reviewer's issue with the term phenotypically-defined LSC, we have re-adjusted this term to the more specific term immunophenotypically-defined LSC, an extensively used term to define a population of cells that share the same cell surface markers (Pollyea & Jordan, Blood 2017). Concerning the second question, we would have to transplant purified immunophenotypically-defined LSC populations into immunocompromised mice and assess AML establishment in order to confirm that (some of) these are not representing healthy mono/ myeloblasts instead. Since this was not feasible, we compromised by using primary AML samples that all have over 80% blast cells (as determined by the extensive immunophenotyping performed at the Hematology Service of the Portuguese Institute of Oncology), thus lowering (but not completely eliminating) the risk that (some of) these CD33+ CD38- CD34+ cells are myeloblasts instead of LSCs. This specification has been added to the revised **Materials and Methods (page 18)**.

6. *The improvement of doxo/cyta/DAC over doxo/cyta in patient samples only really looks convincing for 2/8 cases.*

Considering that each condition shows different degree of survival, we believe that displaying the absolute numbers instead of relative frequencies of CD33+ CD38- CD34+ LSCs is a more accurate depiction of the therapeutic effects of the different regimens. To strengthen the data supporting our claim, we have therefore updated **Figure 6g** panel in which we quantify the absolute number (instead of relative frequency) of immunophenotypically-defined LSCs 6 days after treatment in NT, Doxo+Cyta and Doxo+Cyta+DAC groups (described on **page 14, lines 21-25**). Accordingly, we found significant reductions on LSC absolute numbers in 6 out of 8 samples (labeled 1, 3, 12, 15, 20 and 25).

7. *Based on the findings and the narrative, it would be important to target CDKN1C either genetically or pharmacologically and show that this reduces resistance to doxo + cyta.*

To address the reviewer's suggestion we have tried to establish CDKN1C KO HEL and OCIAML3 lines, but unfortunately failed to do so mainly due to unexpected loss of Cas9 activity in these cell lines. This notwithstanding, in the experiments done to address Reviewers' comments, we found that chemoresistance development associates not only with CDKN1C but also with multiple other transcriptomic alterations. In detail: induction of multiple genes associated with quiescence induction (CDKN1B, CDKN2D, EGR1, JUN, TXNIP), induction of ABC genes (ABCA2, B6, C3) and also self-renewal genes (HOXA4, B4, B5, FOXO1, CBX7, etc.). Thus, although we consider CDKN1C manipulation experiments interesting, we do not consider them essential for the narrative in the context of new version of the manuscript as there are likely multiple compensations operating upon CDKN1C depletion – an hypothesis that we instead intend to address in follow-up studies.

8. *There are several large, publicly available AML datasets (TCGA, Beat AML, etc.). It would be useful to look at CDKN1C expression versus mutational/ prognostic groups in these datasets.*

As suggested, we have looked for CDKN1C expression levels in a relevant AML dataset (<http://servers.binf.ku.dk/bloodspot/>). We observed that CDKN1C is significantly overexpressed in AML with translocations t(15;17) compared to normal karyotype AML. Conversely CDKN1C is very low expressed in ALL samples (see figure below). We have not added this analysis to the paper since these are publicly available data and our current manuscript provides many original findings of broader scope (as highlighted in our reply to point #7).

9. *It is often not clear which cell line is used for data in the figures, particularly in the main paper figures after Fig 1. It seems as though much of the data was generated from a single cell line (HEL), though some of it may have been replicated in OCI-AML3 in the supplement. It will be important to be very clear which line is being used for every experiment and finding. For instance, for down-regulated CDKN1C with doxo/cyta/DAC, has this been validated in both cell lines or only in HEL?*

The identification of which cell line is being used for each specific figure is now clearly indicated in the figure legend of each main and supplementary figure of the revised manuscript. CDKN1C expression was assessed in HEL cell line (**Figure 4**) and validated by qPCR in OCIAML3 cell line (**new Supplementary Figure 9a**).

REVIEWERS' COMMENTS:

Reviewer #1 (Remarks to the Author):

I might be not entirely happy with all answers. However, I have to admit, authors showed considerable effort to improve the readability of the manuscript and clear controversial points. I am ready to accept all answers to my points of concern. Yet, the reference to the stem cell signature with the reference to the Milanovich et al. 2018 remains confusing and technically incorrect. Milanovich et al. performed different kind of experiments and related it to stress responses and senescence, the signature term was used in the context of adult tissue signature and embryonic stem-cell signature (with the reference to other publications). I would suggest either to avoid it completely or specify more clearly which signature is really implied.

Reviewer #2 (Remarks to the Author):

After reading the revised manuscript I think that the manuscript has been significantly improved and I have no further comments.

Reviewer #3 (Remarks to the Author):

Caiado et al. did a tremendous job addressing all points made by the reviewers. I appreciate the detailed responses and the additional experimental work performed. The revised manuscript has been substantially improved in both clarity and dimension. It will be of great value to basic and translational cancer researchers.

The epigenetic studies reported in the reviewer-only figure do not fully rule out the possibility of an epigenetic mechanism of the observed therapeutic resistance and re-sensitization phenomena due to the limited number of loci tested. At the same time, I do agree with the authors that a more in-depth profiling is outside of scope of the present investigation. Since Caiado et al. dedicate a substantial amount of space to setting the stage for non-genetic mechanisms in the introduction, they should mention potential epigenetic regulation in the discussion.

I have no further comments. Thank you.

Reviewer #4 (Remarks to the Author):

the authors have attempted to address most of the comments from the reviewers. in some cases this has been successful, in others technical issues precluded experiments from being performed to completion. It does appear that the authors have made a good faith effort to address points and the manuscript has been improved.

Second round point-by-point reply to the Reviewers of NCOMMS-19-04936A

(Caiado et al.)

We thank the Reviewers for the constructive criticism and suggestions to improve our manuscript. We have incorporated their suggestions in our manuscript.

Reviewer #1 (remarks to the authors):

I might be not entirely happy with all answers. However, I have to admit, authors showed considerable effort to improve the readability of the manuscript and clear controversial points. I am ready to accept all answers to my points of concern. Yet, the reference to the stem cell signature with the reference to the Milanovich et al. 2018 remains confusing and technically incorrect. Mllanovich et al. performed different kind of experiments and related it to stress responses and senescence, the signature term was used in the context of adult tissue signature and embryonic stem-cell signature (with the reference to other publications). I would suggest either to avoid it completely or specify more clearly which signature is really implied.

In agreement with the reviewer, this citation referred to multiple stem-cell signatures and not to a single one as assumed by the authors. For clarity and scientific accuracy of the manuscript we have removed the citation by Milanovich et al. 2018 from our current manuscript, as suggested by the reviewer.

Reviewer #2 (remarks to the authors):

Caiado et al. did a tremendous job addressing all points made by the reviewers. I appreciate the detailed responses and the additional experimental work performed. The revised manuscript has been substantially improved in both clarity and dimension. It will be of great value to basic and translational cancer researchers.

The epigenetic studies reported in the reviewer-only figure do not fully rule out the possibility of an epigenetic mechanism of the observed therapeutic resistance and re-sensitization phenomena due to the limited number of loci tested. At the same time, I do agree with the authors that a more in-depth profiling is outside of scope of the present investigation. Since Caiado et al. dedicate a substantial amount of space to setting the stage for non-genetic mechanisms in the introduction, they should mention potential epigenetic regulation in the discussion.

I have no further comments. Thank you.

We thank the reviewer for his kind remarks. Concerning his suggestion, we have now incorporated in the discussion section a brief speculative interpretation of the potential epigenetic mechanisms that operate in our system. Given the established (but still debatable) role of DNA methylation at promoter regions in gene transcriptional repression, we speculate that the observed decitabine-associated gene upregulation might result from a direct hypomethylation effect and consequent increase in their transcription (giving the example of MET, which has been shown to be regulated negatively by promoter methylation). Following the same logic we suggest that the down-regulated genes might result from a hypomethylation dependent transcriptional activation of their negative regulators. Although other potential mechanisms could have been associated to decitabine action (e.g. viral mimicry responses by endogenous transcripts Roulois D et al. 2015) the hypermethylated nature of the AML genome favours the discussed hypothesis.